# Admixture into and within sub-Saharan Africa

**George BJ Busby[1]\*, Gavin Band[1,2], Quang Si Le[1], Muminatou Jallow[3,4], Edith Bougama[5], Valentina D Mangano[6], Lucas N Amenga-Etego[7], Anthony Enimil[8], Tobias Apinjoh[9], Carolyne M Ndila[10], Alphaxard Manjurano[11,12], Vysaul Nyirongo[13], Ogobara Doumba[14], Kirk A Rockett[1,2], Dominic P Kwiatkowski[1,2], Chris CA Spencer[1]\*, Malaria Genomic Epidemiology Network[1,2]**

[1]Wellcome Trust Centre for Human Genetics, Oxford, United Kingdom; [2]Wellcome Trust Sanger Institute, Cambridge, United Kingdom; [3]Medical Research Council Unit, Serrekunda, The Gambia; [4]Royal Victoria Teaching Hospital, Banjul, The Gambia; [5]Centre National de Recherche et de Formation sur le Paludisme, Ouagadougou, Burkina Faso; [6]Dipartimento di Sanita Publica e Malattie Infettive, University of Rome La Sapienza, Rome, Italy; [7]Navrongo Health Research Centre, Navrongo, Ghana; [8]Komfo Anokye Teaching Hospital, Kumasi, Ghana; [9]Department of Biochemistry and Molecular Biology, University of Buea, Buea, Cameroon; [10]KEMRI-Wellcome Trust Research Programme, Kilifi, Kenya; [11]Joint Malaria Programme, Kilimanjaro Christian Medical College, Moshi, Tanzania; [12]Faculty of Infectious and Tropical Diseases, London School of Hygiene and Tropical Medicine, London, United Kingdom; [13]Malawi-Liverpool Wellcome Trust Clinical Research Programme, College of Medicine, University of Malawi, Blantyre, Malawi; [14]Malaria Research and Training Centre, University of Bamako, Bamako, Mali

**\*For correspondence:** george@well.ox.ac.uk (GBJB); spencer@well.ox.ac.uk (CCAS)

**Group author details:** Malaria Genomic Epidemiology Network See page 34

**Competing interests:** The authors declare that no competing interests exist.

**Abstract** Similarity between two individuals in the combination of genetic markers along their chromosomes indicates shared ancestry and can be used to identify historical connections between different population groups due to admixture. We use a genome-wide, haplotype-based, analysis to characterise the structure of genetic diversity and gene-flow in a collection of 48 sub-Saharan African groups. We show that coastal populations experienced an influx of Eurasian haplotypes over the last 7000 years, and that Eastern and Southern Niger-Congo speaking groups share ancestry with Central West Africans as a result of recent population expansions. In fact, most sub-Saharan populations share ancestry with groups from outside of their current geographic region as a result of gene-flow within the last 4000 years. Our in-depth analysis provides insight into haplotype sharing across different ethno-linguistic groups and the recent movement of alleles into new environments, both of which are relevant to studies of genetic epidemiology.

## Introduction

Advances in DNA analysis technology and the drive to understand the genetic basis of human phenotypes has led to a rapid growth in the amount of genomic data that is available for analysis. Whilst tens of thousands of genetic variants have been associated with different diseases in populations of European descent (**Welter et al., 2014**), less progress has been made in studies of important diseases in Africa (**Need and Goldstein, 2009**). Several consortia are beginning to focus on understanding the genetic basis of infectious and non-communicable disease specifically in Africa

**eLife digest** Our genomes contain a record of historical events. This is because when groups of people are separated for generations, the DNA sequence in the two groups' genomes will change in different ways. Looking at the differences in the genomes of people from the same population can help researchers to understand and reconstruct the historical interactions that brought their ancestors together. The mixing of two populations that were previously separate is known as admixture.

Africa as a continent has few written records of its history. This means that it is somewhat unknown which important movements of people in the past generated the populations found in modern-day Africa. Busby et al. have now attempted to use DNA to look into this and reconstruct the last 4000 years of genetic history in African populations.

As has been shown in other regions of the world, the new analysis showed that all African populations are the result of historical admixture events. However, Busby et al. could characterize these events to unprecedented level of detail. For example, multiple ethnic groups from The Gambia and Mali all show signs of sharing the same set of ancestors from West Africa, Europe and Asia who mixed around 2000 years ago. Evidence of a migration of people from Central West Africa, known as the Bantu expansion, could also be detected, and was shown to carry genes to the south and east. An important next step will be to now look at the consequences of the observed gene-flow, and ask if it has contributed to spreading beneficial, or detrimental, mutations around Africa.

(*Malaria Genomic Epidemiology Network, 2008*; *2015*; *H3Africa Consortium, 2014*; *Gurdasani et al., 2014*), and a number of recent studies have described patterns of genetic variation across the continent (*Campbell and Tishkoff, 2008*; *Tishkoff et al., 2009*; *Gurdasani et al., 2014*). Analyses of the structure of genetic variation are important in the design, analysis, and interpretation of genetic epidemiology studies – which aim to uncover novel relationships between genes, the environment, and disease (*Malaria Genomic Epidemiology Network, 2015*) – and provide an opportunity to relate patterns of association to historical connections between different human populations.

Admixture occurs when genetically differentiated ancestral groups come together and mix, a process which is increasingly regarded as a common feature of human populations across the globe (*Patterson et al., 2012*; *Hellenthal et al., 2014*; *Busby et al., 2015*). Genome-wide analyses of African populations are refining previous models of the continent's history and its impact on genetic diversity. One insight is the identification of clear, but complex, evidence for the movement of Eurasian ancestry back into the continent as a result of admixture over a variety of timescales (*Pagani et al., 2012*; *Pickrell et al., 2014*; *Gurdasani et al., 2014*; *Hodgson et al., 2014a*; *Llorente et al., 2015*). On a broad sample of 18 ethnic groups from eight countries, the African Genome Variation Project (AGVP) (*Gurdasani et al., 2014*) recreated a previous analysis to identify recent Eurasian admixture, within the last 1.5 thousand years (ky), in the Fulani of West Africa (*Tishkoff et al., 2009*; *Henn et al., 2012*) and several East African groups from Kenya; older Eurasian ancestry (2–5 ky) in Ethiopian groups, consistent with previous studies of similar populations (*Pagani et al., 2012*; *Pickrell et al., 2014*); and a novel signal of ancient (>7.5 ky) Eurasian admixture in the Yoruba of Central West Africa (*Gurdasani et al., 2014*). Comparisons of contemporary sub-Saharan African populations with the first ancient genome from within Africa, a 4.5 ky Ethiopian individual (*Llorente et al., 2015*), provide additional support for limited migration of Eurasian ancestry back into East Africa within the last 3000 years.

Within this timescale, the major demographic change within Africa was the transition from hunting and gathering to pastoralist and agricultural lifestyles (*Diamond and Bellwood, 2003*; *Smith, 2005*; *Barham and Mitchell, 2008*; *Li et al., 2014*). This shift was long and complex and occurred at different speeds, instigating contrasting interactions between the agriculturalist pioneers and the inhabitant people (*Mitchell, 2002*; *Marks et al., 2014*). The change was initialised by the spread of pastoralism (i.e. the raising and herding of livestock) across Africa and the subsequent movement east and south from Central West Africa of agricultural technology together with the

branch of Niger-Congo languages known as Bantu (*Mitchell, 2002*; *Barham and Mitchell, 2008*). The extent to which this cultural expansion was accompanied by people is an active research question, but an increasing number of molecular studies indicate that the expansion of languages was accompanied by the diffusion of people (*Beleza et al., 2005*; *Berniell-Lee et al., 2009*; *Tishkoff et al., 2009*; *Pakendorf et al., 2011*; *de Filippo et al., 2012*; *Ansari Pour et al., 2013*; *Li et al., 2014*; *González-Santos et al., 2015*).

The AGVP also found evidence of widespread hunter-gatherer ancestry in African populations, including ancient (9 ky) Khoesan ancestry in the Igbo from Nigeria, and more recent hunter-gatherer ancestry in eastern (2.5–4.5 ky) and southern (0.9–4 ky) African populations (*Gurdasani et al., 2014*). The identification of hunter-gatherer ancestry in non-hunter-gatherer populations together with the timing of these latter events is consistent with the known expansion of Bantu languages across Africa within the last 3 ky (*Mitchell, 2002*; *Diamond and Bellwood, 2003*; *Smith, 2005*; *Barham and Mitchell, 2008*; *Marks et al., 2014*; *Li et al., 2014*). These studies have described the novel and important influence of both Eurasian and hunter-gatherer ancestry on the population genetic history of sub-Saharan Africa and provide an important description of the movement of alleles and haplotypes into and within the continent, but questions remain of the extent and timing of key events, and their impact on contemporary populations.

Here we analyse genome-wide data from 12 Eurasian and 46 sub-Saharan African groups. Half (23) of the African groups represent subsets of samples collected from nine countries as part of the MalariaGEN consortium. Details on the recruitment of samples in relation to studying malaria genetics are published elsewhere (*Malaria Genomic Epidemiology Network, 2014*; *2015*). The remaining 23 groups are from publicly available datasets from a further eight sub-Saharan African countries (*Pagani et al., 2012*; *Schlebusch et al., 2012*; *Petersen et al., 2013*) and the 1000 Genomes Project (1KGP), with Eurasian groups from the latter included to help understand the genetic contribution from outside of the continent (*Figure 1—figure supplement 1*). With the exception of Austronesian in Madagascar, African languages can be broadly classified into four major macro-families: Afroasiatic, Nilo-Saharan, Niger-Congo, and Khoesan (*Blench, 2006*); and although we have representative groups from each (*Supplementary file 1*), our sample represents a significant proportion of the sub-Saharan population in terms of number, but not does not equate to a complete picture of African ethnic diversity. We created an integrated dataset of genotypes at 328,000 high-quality SNPs and use established approaches for comparing population allele frequencies across groups to provide a baseline view of historical gene-flow. We then apply statistical approaches to phasing genotypes to obtain haplotypes for each individual, and use previously published methods to represent the haplotypes that an individual carries as a mosaic of other haplotypes in the sample (so-called chromosome painting [*Li and Stephens, 2003*]).

We present a detailed picture of haplotype sharing across sub-Saharan Africa using a model-based clustering approach that groups individuals using haplotype information alone. The inferred groups reflect broad-scale geographic patterns. At finer scales, our analysis reveals smaller groups, and often differentiates closely related populations consistent with self-reported ancestry (*Tishkoff et al., 2009*; *Bryc et al., 2010*; *Hodgson et al., 2014a*). We describe these patterns by measuring gene-flow between populations and relate them to potential historical movements of people into and within sub-Saharan Africa. Understanding the extent to which individuals share haplotypes (which we call *coancestry*), rather than independent markers, can provide a rich description of ancestral relationships and population history (*Lawson et al., 2012*; *Leslie et al., 2015*). For each group we use the latest analytical tools to characterise the populations as mixtures of haplotypes and provide estimates for the date of admixture events (*Lawson et al., 2012*; *Hellenthal et al., 2014*; *Leslie et al., 2015*; *Montinaro et al., 2015*). As well as providing a quantitative measure of the coancestry between groups, we identify the dominant events which have shaped current genetic diversity in sub-Saharan Africa. We close by discussing the relevance of these observations to studying genotype-phenotype associations in Africa.

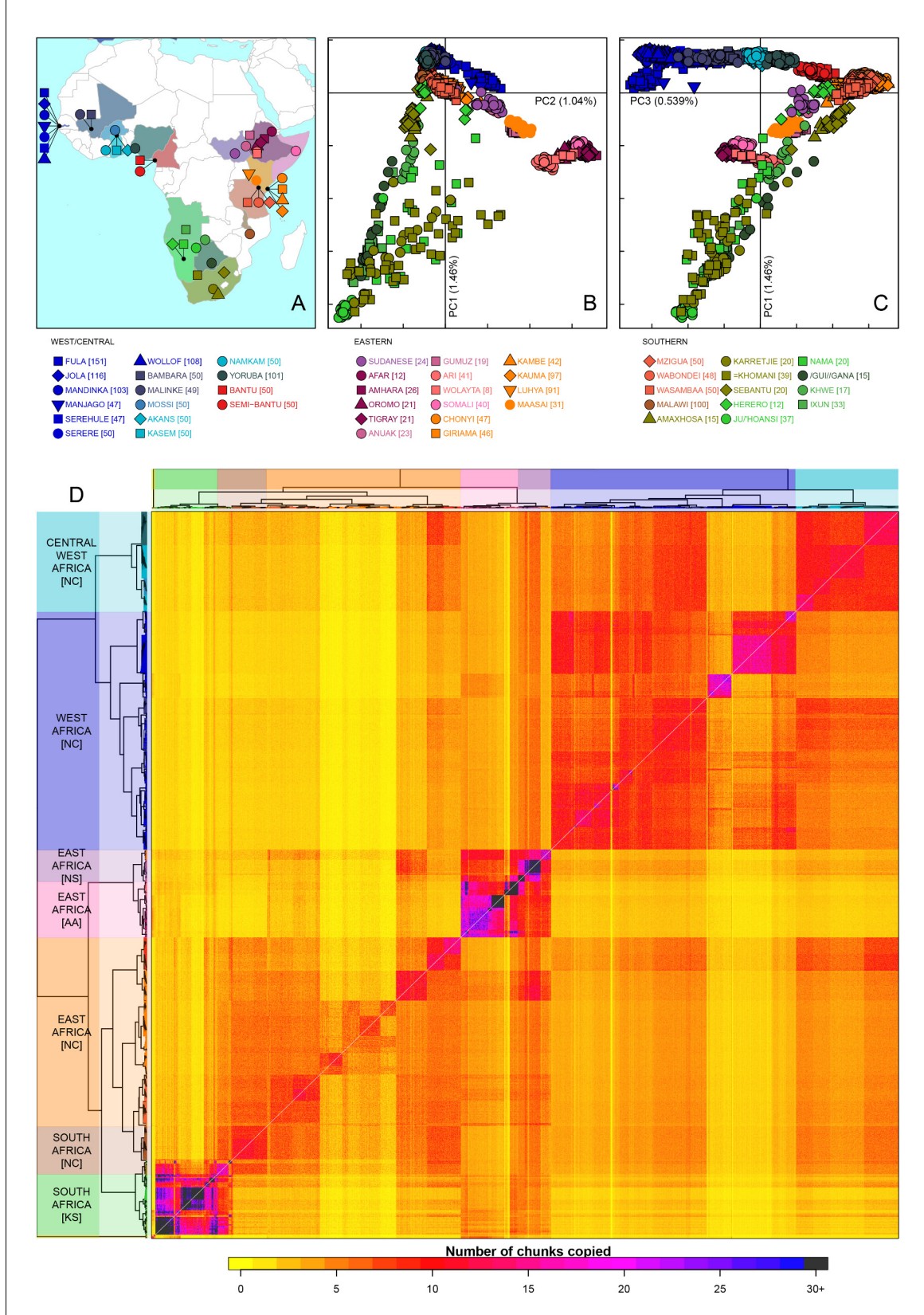

**Figure 1.** Sub-Saharan African genetic variation is shaped by ethno-linguistic and geographical similarity. (**A**) the origin of the 46 African ethnic groups used in the analysis; ethnic groups from the same country are given the same colour, but different shapes; the legend describes the identity of each

*Figure 1 continued on next page*

Figure 1 continued

point. *Figure 1—figure supplement 1* and *Figure 1—source data 1* provide further detail on the provenance of these samples. (B) PCA shows that the first major axis of variation in Africa (PC1, y-axis) splits southern groups from the rest of Africa, each symbol represents an individual; PC2 (x-axis) reflects ethno-linguistic differences, with Niger-Congo speakers split from Afroasiatic and Nilo-Saharan speakers. Tick marks here and in (C) show the scale. (C) The third principle component (PC3, x-axis) represents geographical separation of Niger-Congo speakers, forming a cline from west to east Africans (D) results of the fineSTRUCTURE clustering analysis using copying vectors generated from chromosome painting; each row of the heatmap is a recipient copying vector showing the number of chunks shared between the recipient and every individual as a donor (columns);the tree clusters individuals with similar copying vectors together, such that block-like patterns are observed on the heat map; darker colours on the heatmap represent more haplotype sharing (see text for details); individual tips of the tree are coloured by country of origin, and the seven ancestry regions are identified and labelled to the left of the tree; labels in parentheses describe the major linguistic type of the ethnic groups within: AA = Afroasiatic, KS = Khoesan, NC = Niger-Congo, NS = Nilo-Saharan.

The following source data and figure supplements are available for figure 1:

**Source data 1.** Overview of sampled populations describing the continent, region, numbers of individuals used, and the source of any previously published datasets.
**Figure supplement 1.** Map of populations used in the analysis.
**Figure supplement 2.** An example of hierarchical clustering to chose two groups of similar individuals from the Fula based on a PCA of The Gambia.
**Figure supplement 3.** fineSTRUCTURE analysis of the full dataset.

## Results

### Broad-scale population structure reflects geography and language

Throughout this article we use shorthand current-day geographical and ethno-linguistic labels to describe ancestry. For example we write "Eurasian ancestry in East African Niger-Congo speakers", where the more precise definition would be "ancestry originating from groups currently living in Eurasia in groups currently living in East Africa that speak Niger-Congo languages" (*Pickrell et al., 2014*). We also stress that the use of Khoesan in the current setting refers to groups with shared linguistic characteristics which does not necessarily imply shared close genealogical relationships (*Güldemann and Fehn, 2014*). Our combined dataset included 3283 individuals from 46 sub-Saharan different African ethnic groups and 12 non-African populations (*Figure 1A* and *Figure 1—figure supplement 1*). An initial fineSTRUCTURE analysis (outlined below and in *Figure 1—figure supplement 2* and *Figure 1—figure supplement 3*) demonstrated sub-structure in two of the African ethnic groups, the Fula and Mandinka, so we split both of these populations into two groups, giving a final set of 48 African groups for all analyses.

As an initial description of the genetic structure of the samples we applied principal component analysis to the genotype data (*Patterson et al., 2006*). As in other regions of the world (*Novembre et al., 2008*; *Behar et al., 2010*), the leading principal components show that genetic relationships are broadly defined by geographical and ethno-linguistic similarity (*Figure 1B,C*). The first two principal components (PCs) reflect ethno-linguistic divides: PC1 splits southern Khoesan speaking populations from the rest of Africa, and PC2 splits the East African Afroasiatic and Nilo-Saharan speakers from sub-Saharan African Niger-Congo speakers. The third axis of variation defines east versus west Africa, suggesting that in general, population structure in Africa largely mirrors linguistic and geographic similarity (*Tishkoff et al., 2009*).

To access the information from the combination of markers along chromosomes we phased the genotype data into haplotypes, and applied a previously published implementation of chromosome painting (CHROMOPAINTER [*Lawson et al., 2012*]), to estimate the amount of an individual's genome that is shared with each other individual in the data. More specifically, we paint each recipient individual's genome as a mosaic of haplotype segments (chunks) copied from each other donor individual, and summarise these as *copying vectors*. We used the clustering algorithm implemented in fineSTRUCTURE (*Lawson et al., 2012*) to group individuals purely on the similarity of these copying vectors (*Figure 1* and *Figure 1—figure supplement 3*). The pairwise coancestry between individuals can be visualised as a heatmap with each row being the copying vector for each sample

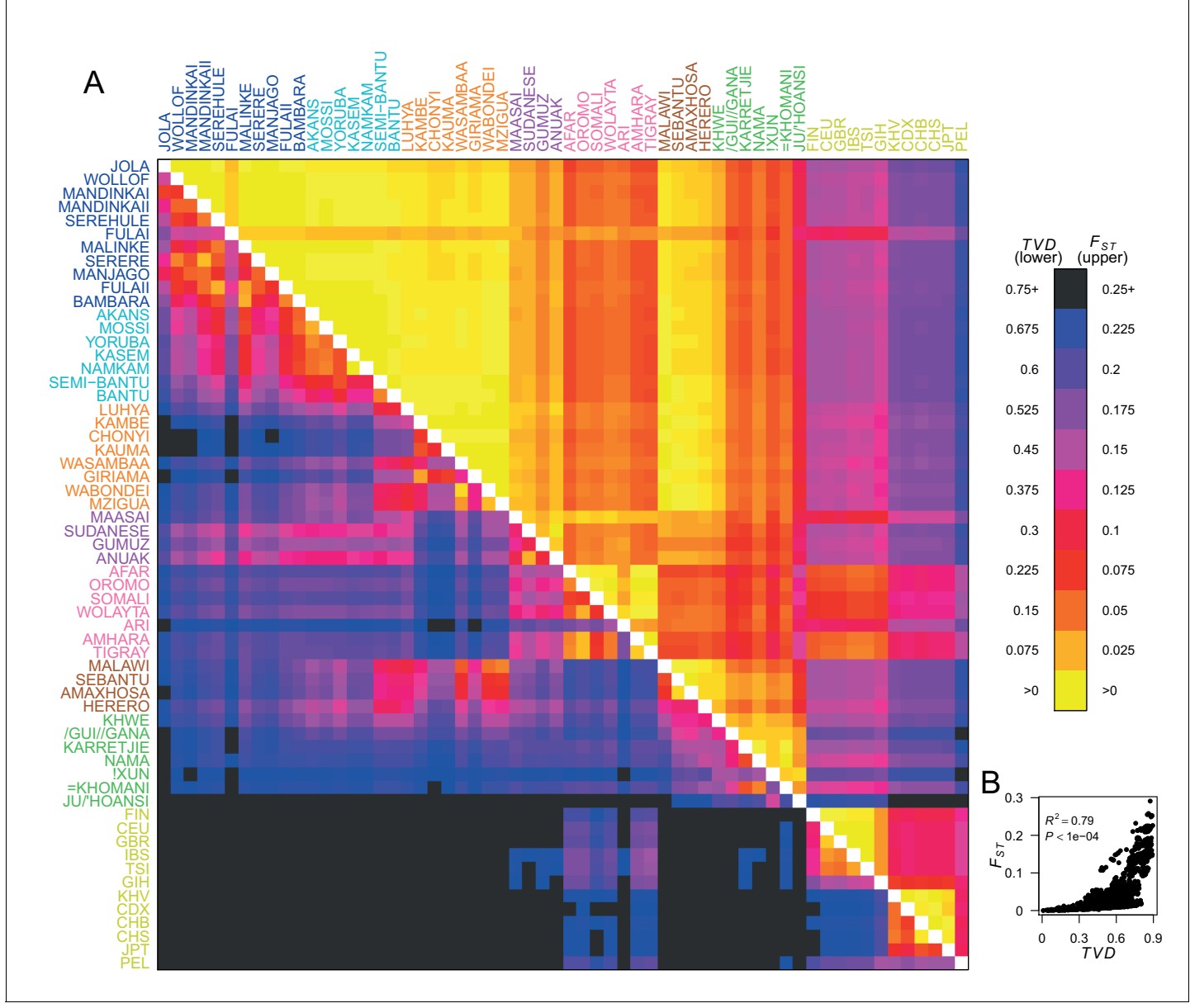

**Figure 2.** Haplotypes capture more population structure than independent loci. (A) For each population pair, we estimated pairwise $F_{ST}$ (upper right triangle) using 328,000 independent SNPs, and *TVD* (lower left triangle) using population averaged copying vectors from CHROMOPAINTER. *TVD* measures the difference between two copying vectors. (B) Comparison of pairwise $F_{ST}$ and *TVD* shows that they are not linearly related: some population pairs have low $F_{ST}$ and high *TVD*. (Source data is detailed in *Figure 2—source data 2* to *Figure 2—source data 1*).

The following source data and figure supplement are available for figure 2:

**Source data 1.** Pairwise *TVD* for Eurasian populations.
**Source data 2.** Pairwise $F_{ST}$ for Eurasian populations.
**Source data 3.** Pairwise $F_{ST}$ for African populations.
**Source data 4.** Pairwise *TVD* for African populations.
**Figure supplement 1.** Haplotypic analysis of populations from the Central West Africa ancestry region accesses fine-scale population differentiation.

(*Figure 1D*), and these are clustered hierarchically to form a tree which describes the inferred relationship between different groups (*Figure 1—figure supplement 3*).

The fineSTRUCTURE analysis identified 154 clusters of individuals, grouped on the basis of copying vector similarity (*Figure 1—figure supplement 3*). Some ethnic groups, such as the Yoruba, Mossi, Jola and Ju/'hoansi form clusters containing only individuals from their own ethnic group. In other populations, most notably from The Gambia and Kenya, individuals from several different ethnic groups cluster together. These are the two countries where the most ethnic groups were sampled, seven and four respectively, and differential sampling could partly explain this observation. Consistent with PCA, the fineSTRUCTURE analysis indicates that African populations tend to share more DNA with geographically proximate populations (dark colours on the diagonal; *Figure 1D*). Block-like structures on the diagonal indicate higher levels of haplotype sharing, as measured by the number of chunks copied, within groups. These patterns are strongest in a subset of the Khoesan speaking individuals (eg. the Ju/'hoansi), several groups from the East Africa (Sudanese, Ari, and Somali groups), and the Fulani and Jola from The Gambia.

Using the results of the PCA and fineSTRUCTURE analyses together with ethno-linguistic classifications and geography, we defined seven groups of populations within Africa (*Supplementary file 1*), which we refer to as *ancestry regions* (shown on the left of *Figure 1D*) when describing geneflow across Africa. From this perspective, the heatmap also shows evidence for coancestry across the continent (more chunks copied away from the diagonal), which is indicative of historical connections between modern-day groups. For example, east Africans from Kenya, Malawi and Tanzania tend to share more DNA with west Africans (lower right) than vice versa (upper left), which suggests that more haplotypes may have spread from west to east Africa. These patterns of coancestry provide evidence of widespread sharing of haplotypes within and between ancestry regions.

## Haplotypes reveal subtle population structure

To quantify population structure, we used two metrics to measure the difference between each of the 48 African and 12 Eurasian groups. First, we used the classical measure $F_{ST}$ (*Hudson et al., 1992*; *Bhatia et al., 2013*) which measures the differentiation in SNP allele frequencies between two groups. The second metric uses the difference in copying vectors between two groups to estimate the total variation distance (TVD) (*Leslie et al., 2015*) at the haplotypic level which provides an alternative measure of differentiation based on combinations of alleles at SNPs along chromosomes. *Figure 2A* shows these two metrics side by side in the upper and lower diagonal. When compared to the level of differentiation between Eurasian and African populations, $F_{ST}$ measured at our integrated set of SNPs is relatively low between many groups from West, Central, and East Africa (yellows on the upper right triangle). In contrast, TVD between the same populations highlights haplotypic differences within Africa which are as strong as between Europe and Asia (pink and purples in lower left triangle). Whilst pairwise TVD tends to increase with pairwise $F_{ST}$ the relationship is neither perfect (Pearson's correlation $R^2$ = 0.79) nor linear (*Figure 2B*). For example, the Chonyi from Kenya have relatively low $F_{ST}$ but high TVD with West African groups, like the Jola (Chonyi-Jola $F_{ST}$ = 0.019; Chonyi-Jola TVD = 0.803) showing that, whilst allele frequency differences between the two populations are relatively low, when we compare the populations' copying vectors, the haplotypic differences are some of the strongest between sub-Saharan groups.

In *Figure 2—figure supplement 1* we show a comparison of PCA, based on genotype data, and fineSTRUCTURE, which uses haplotypes, from a subset of individuals from the Central West African Niger-Congo ancestry region (from Nigeria, Ghana, and Burkina Faso). Whilst we observe some, limited, population structure with PCA, when we look at the copying vectors, we can see the subtle differences in copying that cause fineSTRUCTURE to separate the five ethnic groups into clusters containing only other individuals from their own ethnic group of individuals. The exception to this are the Namkam and Kasem, who are very genetically similar (pairwise $F_{ST}$ of < 0.001) and are merged into a single group. So, consistent with results in European populations (*Leslie et al., 2015*; *Busby et al., 2015*), chromosome painting analyses of African groups can reveal subtle population structure that is hard to detect using approaches based on genotypes alone (for example PCA and $F_{ST}$). Taken together, these observations motivate using haplotype-based approaches to characterise population relationships, in addition to those which consider allele frequencies on their own.

## Allele frequency differences show widespread evidence for admixture

As argued above, a full analysis of admixture best leverages haplotype structure, and we return to this below. To gain an initial understanding of admixture, we applied previously published approaches which analyse the correlations in allele frequencies within and between populations (*Pickrell et al., 2014*; *Gurdasani et al., 2014*). The first approach, the three-population test ($f_3$ statistic [*Reich et al., 2009*]), estimates the proportion of shared genetic drift between a target population and two potential source populations to identify significant departures from the null model of no admixture. Negative values are indicative of canonical admixture events where the allele frequencies in the target population are intermediate between the two source populations. Consistent with recent research (*Pickrell et al., 2014*; *Pickrell and Reich, 2014*; *Gurdasani et al., 2014*; *Llorente et al., 2015*), the majority (83%, 40/48), but not all, of the African groups surveyed showed evidence of admixture ($f_3 < -5$). (*Supplementary file 2*). We do not infer admixture using this statistic in the Jola, Mossi, Kasem, Namkam, Yoruba, Sudanese, Gumuz, and Ju/'hoansi. In most other groups the most significant $f_3$ statistic includes either the Ju/'hoansi or a 1KGP European source (GBR, CEU, FIN, or TSI). Niger-Congo speaking groups from Central West and Southern Africa tend to show most significant statistics involving the Ju/'hoansi, whereas West and East African and Southern Khoesan speaking groups tended to show most significant statistics involving European sources, consistent with an recent analysis on a similar (albeit smaller) set of African populations (*Gurdasani et al., 2014*).

The second approach, ALDER (*Loh et al., 2013*; *Pickrell et al., 2014*) (*Supplementary file 2*) exploits the fact that correlations between allele frequencies along the genome decay over time as a result of recombination. Linkage disequilibrium (LD) can be generated by admixture events, and leaves detectable signals in the genome that can be used to infer historical processes (*Loh et al., 2013*). Following *Pickrell et al. (2014)* and the AGVP (*Gurdasani et al., 2014*), we computed weighted admixture LD curves using the ALDER (*Loh et al., 2013*) package and the HAPMAP recombination map to characterise the sources and timing of gene-flow events. Specifically, we estimated the y-axis intercept (amplitude) of weighted LD curves for each target population using curves from an analysis where one of the sources was the target population (self reference) and the other was, separately, each of the other (non-self reference) populations. Theory predicts that the amplitude of these 'one-reference' curves becomes larger the more similar the non-self reference population is to the true admixing source (*Loh et al., 2013*). As with the $f_3$ analysis outlined above, for many of the sub-Saharan African populations, Eurasian and hunter-gatherer groups (such as the Ju/'hoansi) produced the largest amplitudes (*Figure 3—figure supplement 1* and *Figure 3—figure supplement 2*), reinforcing the contribution of these ancestries to our broad set of African populations.

We investigated the evidence for more complex admixture using MALDER (*Pickrell et al., 2014*), an implementation of ALDER which fits a mixture of exponentials to weighted LD curves to infer multiple admixture events (*Figure 3* and *Figure 3—source data 1*). In *Figure 3A*, for each target population, we show the ancestry region of the two populations with the greatest MALDER curve amplitudes, together with the date of admixture, for at most two events. Throughout, we convert time since admixture in generations to a date by assuming a generation time of 29 years (*Fenner, 2005*). We note that the inferred admixture dates indicate when gene-flow occurred between populations and not the arrival of groups into an area, which may often be several generations earlier.

In general, we find that groups from similar ancestry regions tend to have inferred admixture events at similar times and involving similar sources (*Figure 3*), which suggests that genetic variation has been shaped by shared historical events. For every event, the curves with the greatest amplitudes involved a population from a (usually non-Khoesan) African ancestry region on one side, and either a Eurasian or Khoesan population on the other. To provide more detail on the composition of the admixture sources, we compared MALDER curve amplitudes using source groups from different ancestry regions (central panel *Figure 3A*). In general, we were unable to precisely define the ancestry of the African source of admixture, as curves involving populations from multiple different ancestry regions were not statistically different from each other ($Z < 2$; SOURCE 1). Conversely, comparisons of MALDER curves when the second source of admixture was Eurasian (dark yellow) or

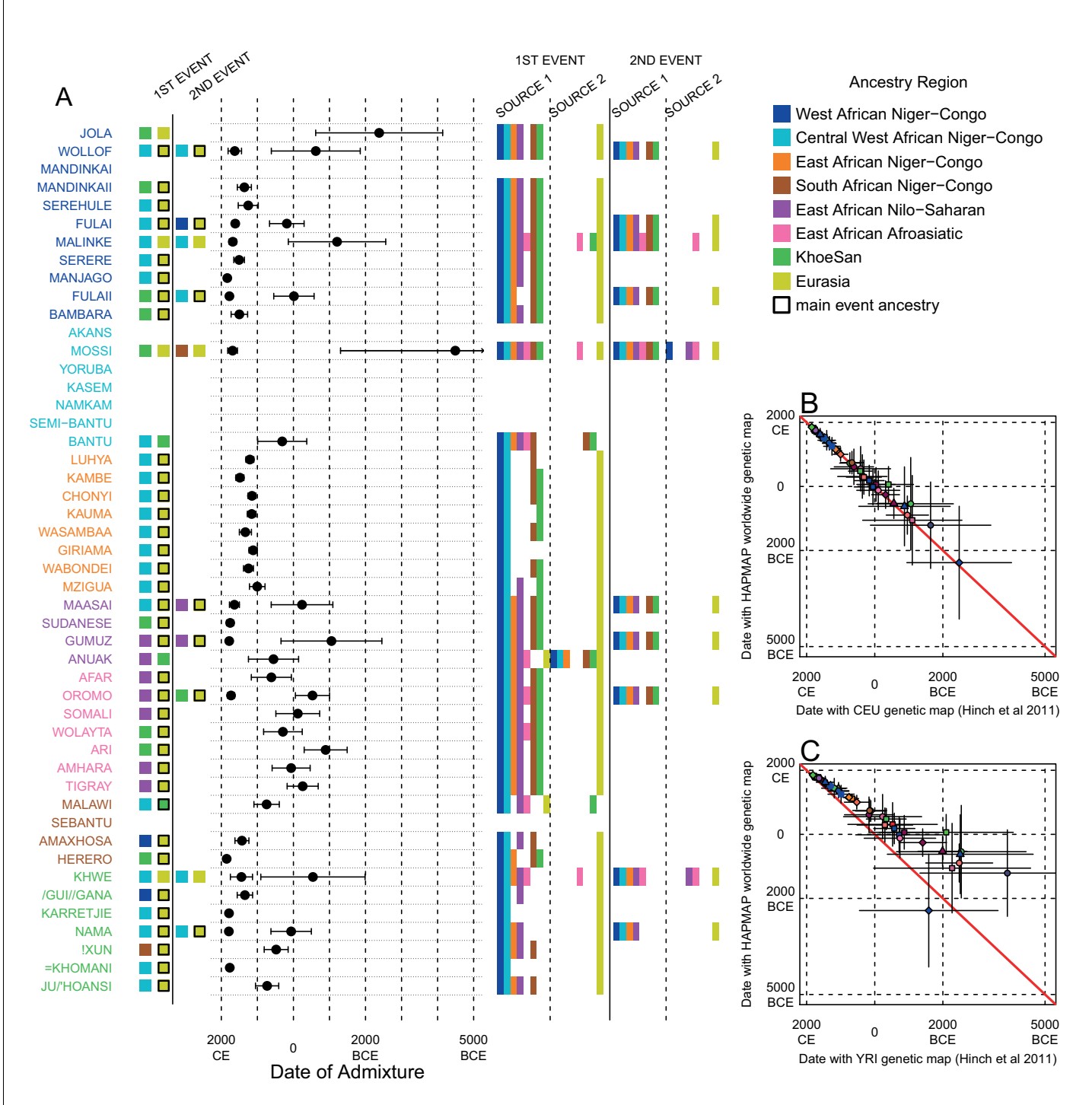

**Figure 3.** Inference of admixture in sub-Saharan Africa using MALDER. We used MALDER to identify the evidence for multiple waves of admixture in each population. (**A**) For each population, we show the ancestry region identity of the two populations involved in generating the MALDER curves with the greatest amplitudes (coloured blocks) for at most two events. The major contributing sources are highlighted with a black box. Populations are ordered by ancestry of the admixture sources and dates estimates which are shown ± 1.96 × s.e. For each event we compared the MALDER curves with the greatest amplitude to other curves involving populations from different ancestry regions. In the central panel, for each source, we highlight the ancestry regions providing curves that are not significantly different from the best curves. In the Jola, for example, this analysis shows that, although the curve with the greatest amplitude is given by Khoesan (green) and Eurasian (dark yellow) populations, curves containing populations from any other African group (apart from Afroasiatic) in place of a Khoesan population are not significantly smaller than this best curve (SOURCE 1). Conversely, when comparing curves where a Eurasian population is substituted with a population from another group, all curve amplitudes are significantly smaller (Z<2).

*Figure 3 continued on next page*

*Figure 3 continued*

(B) Comparison of dates of admixture ± 1.96 × s.e. for MALDER dates inferred using the HAPMAP recombination map and a recombination map inferred from European (CEU) individuals from *Hinch et al. (2011)*. We only show comparisons for dates where the same number of events were inferred using both methods. Point symbols refer to populations and are as in *Figure 1*. (C) as (B) but comparison uses an African (YRI) map. Source data can be found in *Figure 3—source data 1*.

The following source data and figure supplements are available for figure 3:

**Source data 1.** The evidence for multiple waves of admixture in African populations using MALDER and the HAPMAP recombination map.

**Source data 2.** The evidence for multiple waves of admixture in African populations using MALDER and the African recombination map.

**Source data 3.** The evidence for multiple waves of admixture in African populations using MALDER and the European recombination map.

**Source data 4.** The evidence for multiple waves of admixture in African populations using MALDER and the HAPMAP recombination map and a mindis of 0.5cM.

**Figure supplement 1.** Weighted LD amplitudes for a selection of 9 ethnic groups.

**Figure supplement 2.** Comparison of weighted LD amplitude scores across all African ethnic groups.

**Figure supplement 3.** Comparison of the minimum distance to begin computing admixture LD.

**Figure supplement 4.** Comparison of the minimum distance to begin computing admixture LD split by region.

**Figure supplement 5.** Results of MALDER for all populations using a European specific recombination map.

**Figure supplement 6.** Results of the MALDER analysis computing weighted admixture decay curves from 0.5cM.

**Figure supplement 7.** Results of MALDER for all populations using an African specific recombination map.

Khoesan (green), showed that these groups were usually the single best surrogate for the second source of admixture (SOURCE 2).

MALDER uses as input a genetic map to model the expected decay in linkage disequilibirum. We observed a large amount of shared LD at short genetic distances between different African populations (*Figure 3—figure supplement 3* and *Figure 3—figure supplement 4*). Such patterns may result from population genetic processes other than admixture, such as shared demographic history and population bottlenecks (*Loh et al., 2013*). In the main MALDER analysis we present, short-range LD is removed by computing curves at genetic distances <2cM where they are correlated between target and reference population. We provide supplementary analyses where this setting was overridden by allowing MALDER to start computing LD decay curves at short genetic distances (from 0.5cM), irrespective of any short-range correlations in LD between populations. The main difference between the two analyses is that we do not observe previously reported ancient admixture events in Central West African groups (*Gurdasani et al., 2014*) without allowing curves to be computed from 0.5cM. Interpretation of such results is therefore challenging.

Inference of older events relies on modelling the decay of LD over short genetic distances because recombination has had more time to break down correlations in allele frequencies between neighbouring SNPs. We investigated the effect of using European (CEU) and Central West African (YRI) specific recombination maps (*Hinch et al., 2011*) on the dating inference. Whilst dates inferred using the CEU map were consistent with those using the HAPMAP recombination map (*Figure 3B*), when using the African map dates were consistently older (*Figure 3C*), although still generally within the last 7ky. There was also variability in the number of inferred admixture events for some populations between the different map analyses (*Figure 3—figure supplement 5* and *Figure 3—figure supplement 6*).

Many West African groups show evidence of admixture within the last 4 ky involving African and Eurasian sources. The Mossi from Burkina Faso have the oldest inferred date of admixture, at

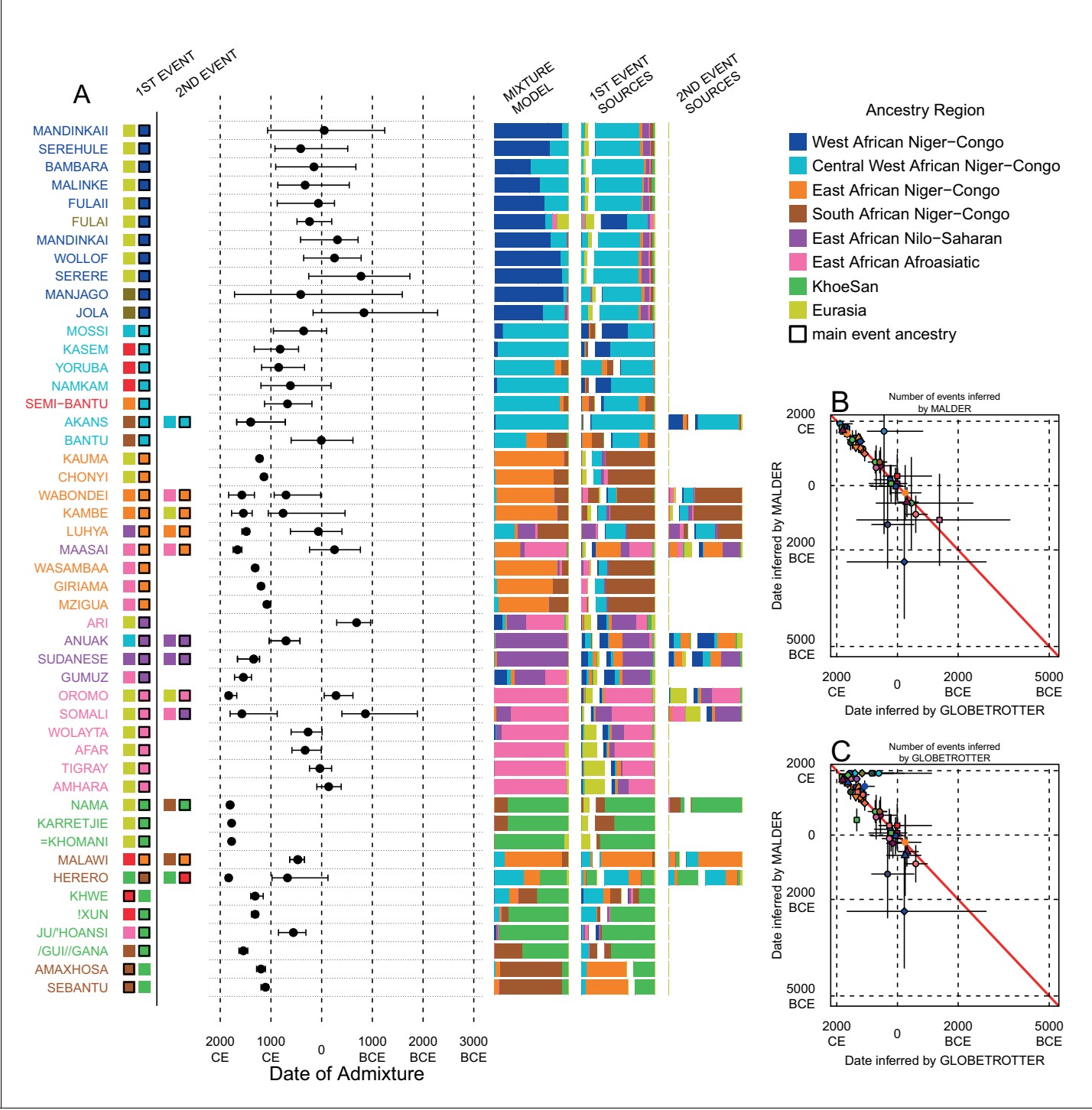

**Figure 4.** Inference of admixture in sub-Saharan African using GLOBETROTTER. (**A**) For each group we show the ancestry region identity of the best matching source for the first and, if applicable, second events. Events involving sources that most closely match FULAI and SEMI-BANTU are highlighted by golden and red colours, respectively. Second events can be either multiway, in which case there is a single date estimate, or two-date in which case 2ND EVENT refers to the earlier event. The point estimate of the admixture date is shown as a black point, with 95% CI shown with lines. MIXTURE MODEL: We infer the ancestry composition of each African group by fitting its copying vector as a mixture of all other population copying vectors. The coefficients of this regression sum to 1 and are coloured by ancestry region. 1ST EVENT SOURCES and 2ND EVENT SOURCES show the ancestry breakdown of the admixture sources inferred by GLOBETROTTER, coloured by ancestry region as in the key top right. (**B**) and (**C**) Comparisons of dates inferred by MALDER and GLOBETROTTER. Because the two methods sometimes inferred different numbers of events, in (**B**) we
*Figure 4 continued on next page*

*Figure 4 continued*

show the comparison based on the inferred number of events in the MALDER analysis, and in (**C**) for the number of events inferred by GLOBETROTTER. Point symbols refer to populations and are as in *Figure 1* and source data can be found in *Figure 4—source data 1*.

The following source data and figure supplements are available for figure 4:

**Source data 1.** Results of the main GLOBETROTTER analysis.

**Source data 2.** Results of the main GLOBETROTTER analysis.

**Figure supplement 1.** Admixture source inference by GLOBETROTTER after sequentially removing local surrogates from the analysis.

**Figure supplement 2.** Admixture source inference by GLOBETROTTER after sequentially removing local surrogates from the analysis.

roughly 5000BCE. Across East Africa Niger-Congo speakers (orange) we infer admixture within the last 4 ky (and often within the last 1 ky) involving Eurasian sources on the one hand, and African sources containing ancestry from other Niger-Congo speaking African groups from the west, on the other. Despite events between African and Eurasian sources appearing older in the Nilo-Saharan and Afroasiatic speakers from East Africa, we see a similar signal of very recent Central West African ancestry in a number of Khoesan groups from Southern Africa, such as the Khwe and /Gui //Gana, together with Malawi-like (brown) sources of ancestry in recent admixture events in East African Niger-Congo speakers.

Most events involved sources where Eurasian (dark yellow in *Figure 3A*) groups gave the largest amplitudes. In considering this observation, it is important to note that the amplitude of LD curves will partly be determined by the extent to which a reference population has differentiated from the target. Due to the genetic drift associated with the out-of-Africa bottleneck and subsequent expansion, Eurasian groups will tend to generate the largest curve amplitudes even if the proportion of this ancestry in the true admixing source is small (*Pickrell et al., 2014*) (in our dataset, the mean pairwise $F_{ST}$ between Eurasian and African populations is 0.157; *Figure 2A* and *Figure 2—source data 2*). To some extent this also applies to Khoesan groups (green in *Figure 3A*), who are also relatively differentiated from other African groups (mean pairwise $F_{ST}$ between Ju/'hoansi and all other African populations in our dataset is 0.095; *Figure 2A* and *Figure 2—source data 2*). In light of this, and the observation that curves involving groups from different ancestry regions are often not different from each other, it is difficult to infer the proportion or nature of the African, Khoesan, or Eurasian admixing sources, only that the sources themselves contained African, Khoesan, or Eurasian ancestry. Moreover, given uncertainty in the dating of admixture when using different maps and MALDER parameters, these results should be taken as a guide to the general structure of genetics relationships between African groups, rather than a precise description of the gene-flow events.

## Modelling gene-flow with haplotypes

Chromosome painting analysis provides an alternative approach to inferring admixture events which directly uses the similarity in haplotypes (combination of alleles) between pairs of individuals. Evidence of haplotype sharing suggests that the ancestors of two individuals must have been geographically proximal at some point in the past, and the distance over which haplotype sharing extends along chromosomes is inversely related to how far in the past coancestry events have occurred.

We can use copying vectors inferred through chromosome painting to help identify those populations that share ancestry with a recipient group by fitting each vector as a mixture of all other population vectors (*Leslie et al., 2015*; *Montinaro et al., 2015*; *van Dorp et al., 2015*). *Figure 4A* shows the contribution that each ancestry region makes to these mixtures (MIXTURE MODEL column). Almost all groups can best be described as mixtures of ancestry from different regions. For example, the copying vector of the Bantu ethnic group from Cameroon is best described as a combination of 40% Central West African Niger-Congo (sky blue), 30% Eastern Niger-Congo (orange), 25% Southern Niger-Congo (brown), and the remaining 5% coming from West African Niger-Congo (dark blue)

and Khoesan-speaking (green) groups. The mixture model approach is useful for describing coancestry between populations which can result from both admixture and shared evolutionary history.

To explicitly test for and characterise admixture we applied GLOBETROTTER (*Hellenthal et al., 2014*) which is an extension of the mixture model approach described above. Admixture inference can be challenging for a number of reasons: the true admixing source population is often not well represented by a single sampled population; admixture could have occurred in several bursts, or over a sustained period of time; and multiple groups may have come together as complex convolution of admixture events. GLOBETROTTER aims to overcome some of these challenges, in part by using painted chromosomes to explicitly model the correlation structure among nearby SNPs, but also by allowing the sources of admixture themselves to be mixed (*Hellenthal et al., 2014*). In addition, the approach has been shown to be relatively insensitive to the genetic map used (*Hellenthal et al., 2014*), and therefore potentially provides a more robust inference of admixture events, the ancestries involved, and their dates. GLOBETROTTER uses the recombination distance between chromosomal chunks of the same ancestry to infer the time since historical admixture has occurred.

Throughout we refer to target populations as *recipients*, any other sampled populations used to describe the recipient population's admixture event(s) as *surrogates*, and populations used to paint both recipient and surrogate populations as *donors*. Including closely related individuals in chromosome painting analyses can cause the resulting painted chromosomes to be dominated by donors from these close genealogical relationships, which can mask signals of admixture in the genome (*Hellenthal et al., 2014*; *van Dorp et al., 2015*). To help ameliorate this, we painted chromosomes for the GLOBETROTTER analysis by using CHROMOPAINTER to paint each individual from a recipient group with the set of donors which did not include individuals from within their own ancestry region. We additionally painted all (59) other surrogate populations with the same set of *non-local* donors, and used these copying vectors, together with the non-local painted chromosomes, to infer admixture. Using this approach, we found evidence of recent admixture in all African populations (*Figure 4A*). To summarise these events, we show the composition of the admixing source groups as barplots for each population coloured by the contribution from each African ancestry region and Eurasia, alongside the inferred date (with confidence interval determined by bootstrapping) and the estimated proportion of admixture (*Figure 4*). For each event we also identify the best matching donor population to the admixture sources.

## Direct and indirect gene-flow from Eurasia back into Africa

Both MALDER and GLOBETROTTER analyses identified Eurasian gene-flow in many but not all African populations (*Figure 4*). In several groups from South Africa, and all from Central West Africa (Ghana, Nigeria, and Cameroon), we infer admixture between groups that are best represented by contemporary populations residing in Africa. As GLOBETROTTER is designed to identify the most recent admixture event(s) (*Hellenthal et al., 2014*), this observation does not rule out gene-flow from Eurasia back into these groups, but does suggest that subsequent movements between African groups were more important in generating current genetic diversity in these groups. We also do not observe Eurasian ancestry in all East African Niger-Congo speakers, instead finding more evidence for coancestry with Afroasiatic speaking groups. As we show later, Afroasiatic populations have a significant amount of ancestry from outside of Africa, so the observation of this ancestry in several African groups identifies a route by which Eurasian ancestry may have indirectly entered the continent (*Pickrell et al., 2014*).

Characterising admixture sources as mixtures allows GLOBETROTTER to infer whether Eurasian haplotypes are likely to have come directly into sub-Saharan Africa – in which case the admixture source will contain only Eurasian surrogates – or whether Eurasian haplotypes were brought indirectly together with sub-Saharan groups. In West African Niger-Congo speakers from The Gambia and Mali, we infer admixture involving minor admixture sources which contain mostly Eurasian (dark yellow) and Central West African (sky blue) ancestry, which most closely match the contemporary copying vectors of northern European populations (CEU and GBR) or the Fulani (FULAI, highlighted in gold in *Figure 4A*). The Fulani, a nomadic pastoralist group found across West Africa, were sampled in The Gambia, at the very western edge of their current range, and have previously reported genetic affinities with Niger-Congo speaking, Sudanic, Saharan, and Eurasian populations (*Tishkoff et al., 2009*; *Henn et al., 2012*), consistent with the results of our mixture model analysis

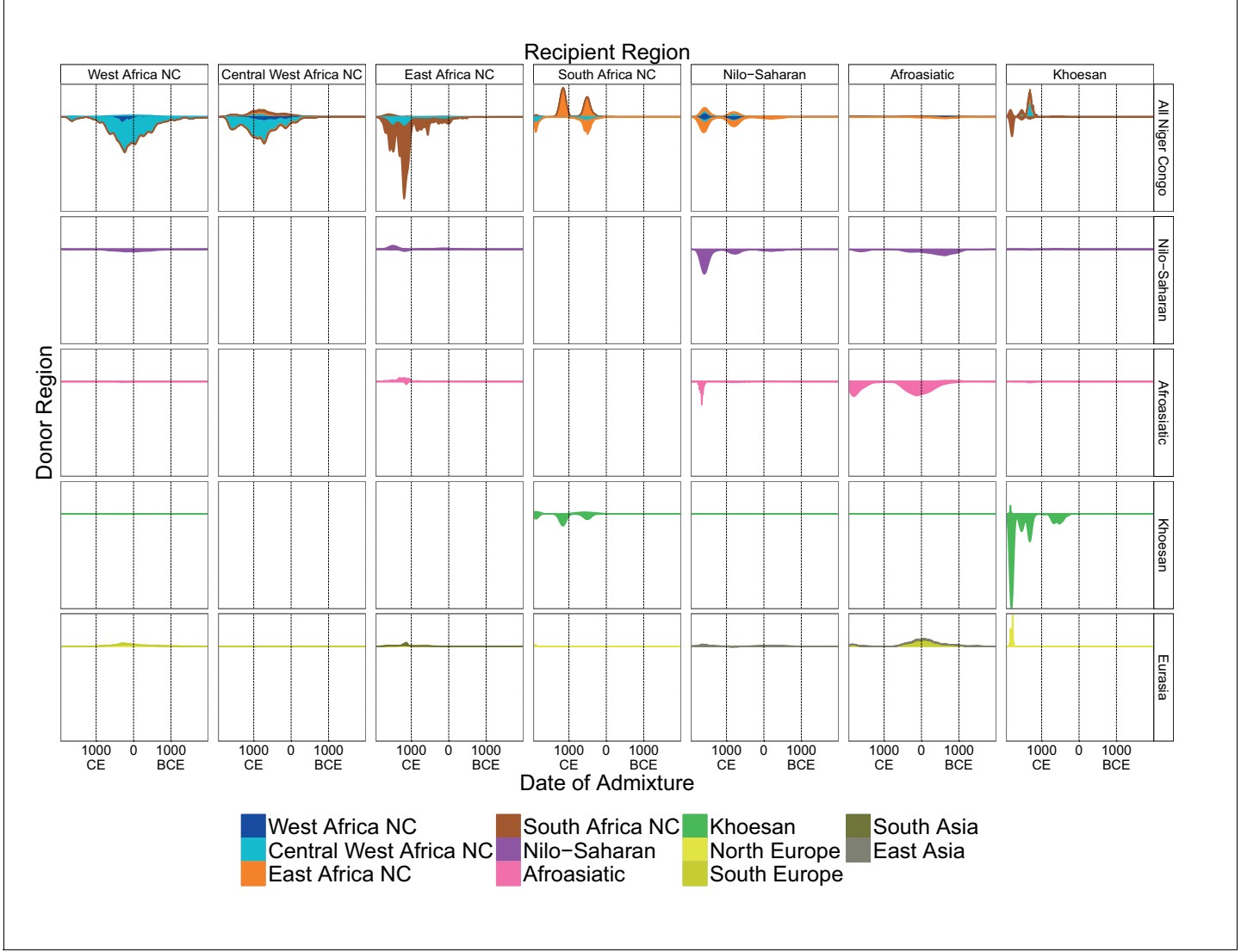

**Figure 5.** A timeline of recent admixture in sub-Saharan Africa. For all events involving recipient groups from each ancestry region (columns) we combine all date bootstrap estimates generated by GLOBETROTTER and show the densities of these dates separately for the minor (above line) and major (below line) sources of admixture. Dates are additionally stratified by the ancestry region of the surrogate populations (rows), with all dates involving Niger Congo speaking regions combined together (All Niger Congo). Within each panel, the densities are coloured by the ancestry region origin of the surrogates, and in proportion to the components of admixture involved in the admixture event. The integrals of the densities are proportional to the admixture proportions of the events contributing to them.

(*Figure 4A*). Admixture in the Fulani differs from other populations from this region, with sources containing greater amounts of Eurasian and Afroasiatic ancestry, but appears to have occurred during roughly the same period (c. 0CE; *Figure 5*).

The Fulani represent the best-matching surrogate to the minor source of recent admixture in the Jola and Manjago, which we interpret as resulting not from specific admixture from them into these groups, but because the mix of African and Eurasian ancestries in contemporary Fulani is the best proxy for the minor sources of admixture in this region. With the exception of the Fulani themselves, the major admixture source in groups across this region is a similar mixture of African ancestries that most closely matches contemporary Gambian and Malian surrogates (Jola, Serere, Serehule, and Malinke), suggesting ancestry from a common West African group within the last 3000 years. The Ghana Empire flourished in West Africa between 300 and 1200CE, and is one of the earliest recorded African states (*Roberts, 2007*). Whilst its origins are uncertain, it is clear that trade in gold,

salt, and slaves across the Sahara, perhaps from as early as the Roman Period, as well as evolving agricultural technologies, were the driving forces behind its development (*Oliver and Fagan, 1975*; *Roberts, 2007*). It is possible these interactions through North Africa, catalysed by trade across the Sahara, allowed gene-flow from Europe and North Africa back into West Africa.

We infer more direct admixture from Eurasian sources in two populations from Kenya, where specifically South Asian populations (GIH, KHV) are the most closely matched surrogates to the minor sources of admixture (*Figure 5*). Interestingly, the Chonyi (1138CE: 1080-1182CE) and Kauma (1225CE: 1167-1254CE) are located on the Kenyan Swahili Coast, a region where Medieval trade across the Indian Ocean is historically documented (*Allen, 1993*), which might explain this Asian admixture. Alternatively, *Blench (2010)* notes that the expansion of Arab shipping down the east Coast of Africa in the 10th Century CE masked the Austronesian (i.e. Oceania and Asia) influence of the resident coastal culture. The implication is that Austronesians, who are known to have contributed genes to Madagascan populations (*Tofanelli et al., 2009*), may also have been in East Africa at about this time. Further work on these groups will help to understand whether the events we observed in the Chonyi and Kauma represent the first evidence of an Austronesian impact in mainland Africa.

In the Kambe, the third group from coastal Kenya, we infer two events, the more recent one involving local groups, and the earlier event involving a European-like source (GBR, 761CE: 461BCE-1053CE). In Tanzanian groups from the same ancestry region, we infer admixture during the same period, this time involving minor admixture sources with Afroasiatic ancestry: in the Giriama (1196CE: 1138-1254CE), Wasambaa (1312CE: 1254-1341CE), and Mzigua (1080: 1007-1138CE). Although the proportions of admixture from these minor sources differ, the major sources of admixture in East African Niger-Congo speakers are similar, containing a mix of Southern Niger-Congo (Malawi), Central West African, Afroasiatic, and Nilo-Saharan ancestries. These events may be an indirect route for European-like gene-flow into East Africa.

In the Afroasiatic speaking populations of East Africa, we infer admixture involving sources containing mostly Eurasian ancestry, which most closely matches the Tuscans (TSI, *Figure 4*). Visualising the temporal distribution of admixture contributions shows that this ancestry appears to have entered the Horn of Africa in two waves (at c. 1800 and 0CE in *Figure 5*) as result of admixture into the Afar (326CE: 7-587CE), Wolayta (268CE: 8BCE-602CE), Tigray (36CE: 196BCE-240CE), and Ari (689BCE:965-297BCE). There are no Middle Eastern groups in our analysis, and this group of events may represent previously observed migrations from the Arabian peninsular at the same time (*Pagani et al., 2012*; *Hodgson et al., 2014a*).

Although Afroasiatic and Nilo-Saharan speakers were sampled from the same part of East Africa, the ancestry of the major sources of admixture of the former do not contain much Nilo-Saharan ancestry and are predominantly Afroasiatic (pink). In Nilo-Saharan speaking groups (purple), the Sudanese (1341CE: 1225–1660), Gumuz (1544CE: 1384–1718), Anuak (703: 427-1037CE), and Maasai (1646CE: 1584-1743CE), we infer greater proportions of West (blue) and East (orange) African Niger-Congo speaking surrogates in the major sources of admixture, indicating both that the Eurasian admixture occurred into groups with mixed Niger-Congo and Nilo-Saharan/Afroasiatic ancestry, and a clear recent link with Central and West African groups.

Lastly, in two Khoesan speaking groups from South Africa, the ≠Khomani and Karretjie, we infer very recent direct admixture involving Eurasian groups most similar to Northern European populations, with dates aligning to European colonial period settlement in Southern Africa (c. 5 generations or 225 years ago; *Figure 5*) (*Hellenthal et al., 2014*). Taken together, and in addition the MALDER analysis above, these observations suggest that gene-flow back into Africa from Eurasia has been common around the edges of the continent, has been sustained over the last 3000 years, and can often be attributed to specific and different historical time periods.

## Population movements within Africa and the Bantu expansion

Before discussing the impact of the Bantu expansion, we highlight three inferred admixture events involving sources unconnected to that migration. We infer admixture in the Ju/'hoansi, a San group from Namibia, involving a source that closely matches a local southern African Khoesan group, the Karretjie, and an East African Afroasiatic, specifically Somali, source at 558CE (311-851CE). Another, older, event in the Maasai (254BCE: 764BCE-239CE) also involves an Afroasiatic source. In contrast the minor source in the event inferred in the Luhya (1486: 1428-1573CE) most closely matches Nilo-

Saharan groups. The recent date of this event implies that Eastern Niger-Congo speaking groups (e.g. the Luhya) interacted with nearby Nilo-Saharan speakers after the putative arrival of Bantu-speaking groups to Eastern Africa which we discuss below.

Most of the sampled groups in this study, and indeed most sub-Saharan Africans, speak a language belonging to the Niger Congo linguistic phylum (*Greenberg, 1972*; *Nurse and Philippson, 2003*). A sub-branch of this group are the so-called 'Bantu' languages – a group of approximately 500 very closely related languages – that are of particular interest because they are spoken by the vast majority of Africans south of the line between Southern Nigeria/Cameroon and Somalia (*Pakendorf et al., 2011*). Given their high similarity and broad geographic range, it is likely that Bantu languages spread across Africa quickly. Bantu languages can themselves be divided into three major groups: northwestern, which are spoken by groups near to the proto-Bantu heartland of Nigeria/Cameroon; western Bantu languages, spoken by groups situated down the west coast of Africa; and eastern, which are spoken across East and Central Africa (*Li et al., 2014*).

Whilst there is linguistic and archaeological consensus that the Bantu heartland was in the general region of southern Nigeria and Cameroon (*Nurse and Philippson, 2003*), it is unclear whether eastern Bantu languages were a primary branch that split off before the western groups began to spread south (the early-split hypothesis), or whether this occurred after the start of the movement south (the late-split hypothesis) (*Pakendorf et al., 2011*). In a study based on glottochronology, *Vansina (1995)* suggests that the expansion started 5kya, whilst estimates based on linguistic diversity are slightly later, around 4kya (*Blench, 2006*). This latter date agrees well with the breakthrough of Neolithic technologies, such as tools and pottery, in the archaeology of the Cameroon proto-Bantu heartlands (*Bostoen, 2007*) and perhaps further south (*Lavachery, 2001*), linking the spread of technology and farming with the Bantu expansion.

The early split hypothesis suggests that the eastern Bantu migrated directly east from Cameroon, 3–2.5 kya (*Nurse and Philippson, 2003*) along the border north of the Congo rainforest, to the Great Lakes Region of East Africa (*Pakendorf et al., 2011*). The late-split hypothesis, on the other hand, suggests that there was an initial spread south, through the equatorial rainforest, with a sub-group splitting east under the rainforest, arriving later in East Africa, potentially around 2kya (*Vansina, 1995*). Regardless of the exact route, the expansion spread south, arriving in southern Africa by the late first millennium CE (*Nurse and Philippson, 2003*). Recent phylogenetic linguistic analysis shows that the relationships between contemporary languages better match predictions based on the late-split hypothesis (*Holden, 2002*; *Currie et al., 2013*; *Grollemund et al., 2015*), an observation supported by genetic analyses (*Li et al., 2014*).

The current dataset does not cover all of Africa. In particular, it contains no hunter-gather groups outside of southern Africa, and no representation of the western Bantu except the Herero from Namibia. Nevertheless, we explored whether our admixture approach could be used to gain insight into the Bantu expansion. Specifically, we wanted to see whether the dates of admixture and composition of admixture sources were consistent with either of the two major models of the Bantu expansion. In the remaining discussion, we make the following assumption: when we observe ancestry from contemporary groups residing in Cameroon (Semi-Bantu and Bantu) this is a proxy for *direct* gene-flow from the origin of the Bantu expansion. Alternatively, higher proportions of ancestry from Southern or Eastern Niger-Congo speakers are the result of subsequent *indirect* gene-flow through these groups, which we use together with the time of admixture to relate to the Bantu expansion. We note that our interpretation may change with future analyses involving populations from the relatively under-sampled central southern Africa.

The major sources of admixture in East African Niger-Congo speakers have both Central West and Southern Niger-Congo ancestry, although it is predominantly the latter (*Figure 4*). If admixture in Eastern Niger-Congo speakers results from early movements directly from Central West Africa (Cameroon surrogates) then we would expect to see sources with predominantly Central West African ancestry. However, all East African Niger-Congo speakers that we sampled have admixture ancestry from a Southern group (Malawi) within the last 2000 years, suggesting that Malawi is more closely related to their Bantu ancestors than Central West Africans on their own. In the SEBantu (1109:1051-1196CE) and AmaXhosa (1196CE: 1109-1283CE), from east southern Africa, we observe reciprocal admixture events involving major sources most similar to East African Niger-Congo speakers. In west southern Africa, on the other hand, we infer two admixture events in the Herero (1834CE: 1805-1892CE and 674CE: 124BCE-979CE), and a single date in the Khoesan-speaking

Khwe (1312; 1152-1399CE), both of which involve sources with higher proportions of ancestry from Cameroon (*Figure 4—source data 1*). In a third west southern African group, the !Xun (1312CE: 1254-1385CE) from Angola, who do not speak a Bantu language, we also infer admixture from a Cameroon-like source at around the same time as the Khwe. The putative Bantu admixture events in Malawi and the Herero occur before those in the !Xun and Khwe (*Figure 4*). This suggests a separate, more recent, arrival for Bantu ancestry in west southern compared to east southern Africa, with the former coming directly down the west coast of Africa and the latter from earlier interactions in central southern Africa (*de Filippo et al., 2012*; *Li et al., 2014*).

To further explore Bantu ancestry in eastern and southern Africa, we performed additional GLOB-ETROTTER analyses where we restricted the surrogate populations used to infer admixture (*Figure 4—figure supplement 1*) to specifically identify ancestry from Cameroon. This analysis allows us to ask whether the indirect Bantu ancestry we observe in East and Southern Niger-Congo speakers can be traced back to the origin of the expansion. When we restrict East African Niger-Congo speakers from having admixture surrogates from either within their ancestry region (locally) or Malawi (*Figure 4—figure supplement 1*), the sources of admixture mainly contained surrogates from the other non-Malawi Southern African Niger-Congo groups (the AmaXhosa, SEBantu, and Herero), reinforcing the relationship between Southern and East African Niger-Congo speakers. With the exception of the Herero, Southern African Niger-Congo speakers show the reverse relationship, choosing East African surrogates when local groups are removed from the inference. Only when both Eastern and Southern African Niger-Congo speakers were restricted from having surrogates from themselves, and each other, did the admixture sources contain significant proportions of Cameroon ancestry (*Figure 4—figure supplement 1*). By contrast, regardless of which surrogates are removed, the Herero always have inferred major admixture sources that contain a majority of Cameroon ancestry (*Figure 4—figure supplement 2*). We discuss the restricted surrogate analysis in further detail in *Supplementary file 3*, *Figure 4—figure supplement 1* and *Figure 4—figure supplement 2*.

In individuals from Malawi we infer a multi-way event with an older date (471: 340-631CE) involving a minor source which mostly contains ancestry from Cameroon, which is, as mentioned, at a similar date to the event seen in the Herero from Namibia. This Bantu admixture appears to have preceded that in other southern Africans by a few hundred years. Given that ancestry from Malawi is often observed in large proportions in the admixture sources of East and Southern African Niger-Congo speakers, and its position between eastern and the most southern groups, Malawi represents the closest proxy in our dataset for the intermediate group that split from the western Bantu. We also see an admixture source in Malawi with a significant proportion of non-Bantu (green) ancestry (2nd event, minor source in *Figure 4*), ancestry which we do not observe in the mixture model analysis, but which is also evident in the other east Southern Niger-Congo speakers (the AmaXhosa and SEBantu) implying that gene-flow must have occurred between the expanding Bantus and the resident hunter-gatherer groups (*Marks et al., 2014*).

In summary, the early date of Bantu admixture in Malawi, its presence as an admixture surrogate across eastern and southern Africa, and the observation of later direct Central West African (Bantu) admixture in western south African groups, highlight the complex dynamics, and multiple waves of migration associated with the movement of Bantu agriculturists from the region around Cameroon into southern and eastern Africa. Moreover, our analysis – in addition to evidence from linguistic phylogenetics (*Currie et al., 2013*; *Grollemund et al., 2015*) – provides genetic support for the late-split hypothesis, suggesting that the agriculturist Bantus migrated south around the Congo rainforest before travelling east.

## A haplotype-based model of gene-flow in sub-Saharan Africa

Our haplotype-based analyses support a complex picture of recent historical gene-flow in Africa (*Figure 6*). Using genetics to infer historical demography will always depend on the available samples and methods used to infer population relationships. Our aim here is to highlight the key gene-flow events that chromosome painting allows us to detect, and to describe their affect on the structure of coancestry:

1. **Colonial Era European admixture in the Khoesan**. In two southern African Khoesan groups we see very recent admixture, within the last 250 years, involving northern European ancestry

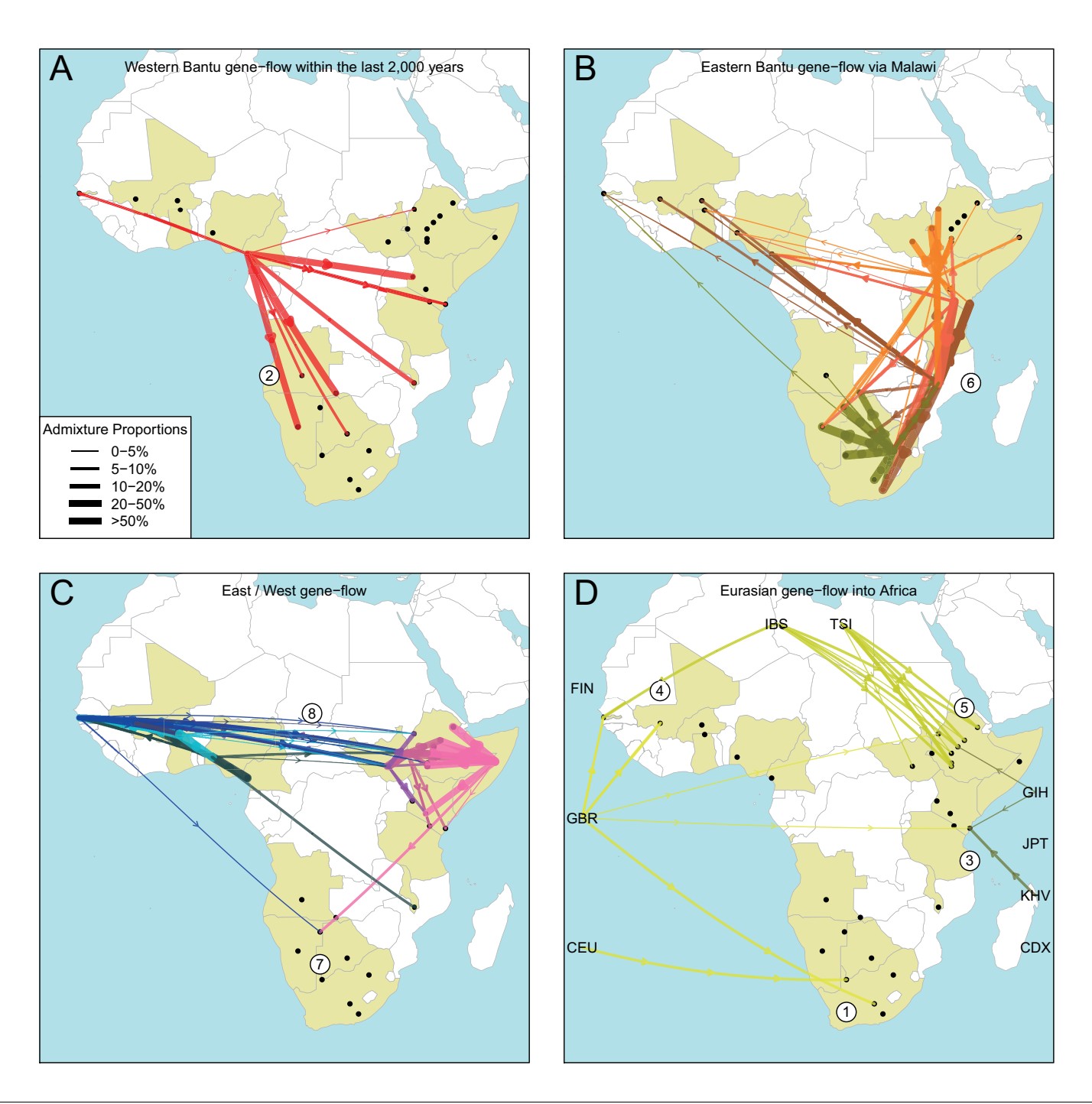

**Figure 6.** The geography of recent gene-flow in Africa. We summarise gene-flow events in Africa using the results of the GLOBETROTTER analysis. For each ethnic group, we inferred the composition of the admixture sources, and link recipient population to surrogates using arrows, the width of which is proportional to the amount it contributes to the admixture event. We separately plot (**A**) all events involving admixture source components from the Bantu and Semi-Bantu ethnic groups in Cameroon; (**B**) all events involving admixture sources from East and Southern African Niger-Congo speaking groups; (**C**) events involving admixture sources from West African Niger-Congo and East African Nilo-Saharan / Afroasiatic groups; (**D**) all events involving components from Eurasia. in (**D**) arrows are linked to the labelled 1KGP Eurasian groups. Arrows are coloured by country of origin, as in *Figure 1—figure supplement 1*. Numbers 1–8 in circles represent the events highlighted in section *A haplotype-based model of gene-flow in sub-Saharan Africa*. An alternative version of this plot, stratified by date, is shown in *Figure 6—figure supplement 1*.
*Figure 6 continued on next page*

*Figure 6 continued*

The following figure supplement is available for figure 6:

**Figure supplement 1.** Gene-flow in Africa over the last 2000 years.

which likely resulted from Colonial Era movements from the UK, Germany, and the Netherlands into South Africa (*Thompson, 2001*).

2. **The recent arrival of the Western Bantu expansion in southern Africa**. Central West African, and in particular ancestry from Cameroon (red ancestry in *Figure 6A*), is seen in Southern African Niger-Congo and Khoesan speaking groups, the Herero, Khwe and !Xun, indicating that the gradual diffusion of Bantu ancestry reached the south of the continent only within the last 750 years. Bantu ancestry in Malawi appears prior to this event.

3. **Medieval contact between Asia and the East African Swahili Coast**. Specific Asian geneflow is observed into two coastal Kenyan groups, the Kauma and Chonyi, which represents a distinct route of Eurasian, in this case Asian, ancestry into Africa, perhaps as a result of Medieval trade networks between Asia and the Swahili Coast around 1200CE.

4. **Gene-flow across the Sahara**. Over the last 3000 years, admixture involving sources containing northern European ancestry is seen on the Western periphery of Africa, in The Gambia and Mali. This ancestry in West Africa is likely to be the result of more gradual diffusion of DNA across the Sahara from northern Africa and across the Iberian peninsular, and not via the Middle East, as in the latter scenario we would expect to see Spanish (IBS) and Italian (TSI) in the admixture sources. We do see limited southern European ancestry in West Africa (*Figures 5* and *6D*) in the Fulani, suggesting that some Eurasian ancestry may also have entered West Africa via North East Africa (*Henn et al., 2012*).

5. **Several waves of Mediterranean / Middle Eastern ancestry into north-east Africa**. We observe southern European gene-flow into East African Afroasiatic speakers over a more prolonged time period over the last 3000 years, with a major wave 2000 years ago (*Figures 5* and *6D*). We do not have Middle-Eastern groups in our analysis, so the observed Italian ancestry in the minor sources of admixture – the Tuscans are the closest Eurasian group to the Middle East – is consistent with previous results using the same samples (*Pagani et al., 2012*; *Hodgson et al., 2014a*), indicating this region as a major route for the back migration of Eurasian DNA into sub-Saharan Africa (*Pagani et al., 2012*; *Pickrell et al., 2014*).

6. **The late split of the Eastern Bantus**. Admixture in East African Niger-Congo speakers occurs during the period 500-1500CE, with a peak around 1000CE. The major sources of admixture in these groups is consistently a mixture of Central West African and Southern Niger-Congo speaking groups, in particular Malawi. This result supports the hypothesis that Bantu speakers initially spread south along the western side of the Congo rainforest before splitting off eastwards, and interacting with local groups in central south Africa – for which Malawi is our best proxy – and then moving further north-east and south (*Figure 6B*).

7. **Pre-Bantu pastoralist movements from East to South Africa**. In the Ju/'hoansi we infer an admixture event involving an East African Afroasiatic source which we date to 311-851CE. This event precedes the arrival of Bantu-speaking groups in southern Africa, and is consistent with several recent results linking east to south Africa and the limited spread of cattle pastoralism prior to the Bantu expansion (*Figures 5* and *6C*) (*Pickrell et al., 2014*; *Ranciaro et al., 2014*; *Macholdt et al., 2015*; *Barham and Mitchell, 2008*).

8. **Ancestral connections between West Africans and the Sudan**. Concentrating on older events, we observe old 'Sudanese' (Nilotic) components in very small proportions in events The Gambia dating to c.0CE (*Figure 4—figure supplement 1* and *Figure 5*) and which may represent ancient expansion relationships between East and West Africa. When we infer admixture in West and Central West African groups without allowing any West Africans to contribute to the inference, we observe a clear signal of Nilo-Saharan ancestry in these groups, consistent with bidirectional movements across the Sahel (*Tishkoff et al., 2009*) and coancestry with (unsampled) Nilo-Saharan groups in Central West Africa. Indeed, if we look again at the PCA in *Figure 1C*, we observe that the Nilo-Saharan speakers are between West and East African Niger-Congo speaking individuals on PC3, an affinity which is supported by the presence of West African components in non-Niger-Congo speaking East Africans (*Figure 6C*).

9. **Ancient Eurasian gene-flow back into Africa and shared hunter-gatherer ancestry**. The $f_3$ statistics show the general presence of ancient Eurasian and/or Khoesan ancestry across much of sub-Saharan Africa. We tentatively interpret these results as being consistent with recent

research suggesting very old (>10 kya) migrations back into Africa from Eurasia (*Hodgson et al., 2014a*), with the ubiquitous hunter-gatherer ancestry across the continent possibly related to the inhabitant populations present across Africa prior to these more recent movements. Future research involving ancient DNA from multiple African populations will help to further characterise these observations.

## Discussion

We have presented an in-depth analysis of the genetic history of sub-Saharan Africa in order to characterise its impact on present day diversity. We show that gene-flow has taken place over a variety of different time scales which suggests that, rather than being static, populations have been sharing DNA, particularly over the last 3000 years. An important question in African history is how contemporary populations relate to those present in Africa before the transition to pastoralism that began in the Nile Valley some 9kya. The $f_3$ and MALDER analyses show evidence for deep Eurasian and some hunter-gatherer ancestry across Africa, to which our GLOBETROTTER analysis (*Figure 4*) provides further clarity on the composition of the admixture sources, as well as the timing of events and their impact on groups in our analysis (*Figure 6*). On the basis of our analysis, none of the African populations in our study has remained isolated and unchanged over the last 4000 years.

With a couple of exceptions (some of the events we have highlighted in *Figure 6*), the major signals of admixture in our analysis relate to the movement of Eurasian ancestry back into Africa and the movement of genes south and east from Central West Africa, likely as a result of the Bantu expansion. The transition from foraging to pastoralism and agriculture in Africa is likely to have been complex, with its impact on existing populations varying substantially. Our analysis provides an estimate of the timing of this expansion (*Figure 5*). It is important to note that dates of admixture inferred through genetics will always be more recent than the date at which two populations have come together. Our dataset is not an exhaustive sample of African populations, and there are likely to be other events than those reported here that have been important in generating the current genetic landscape of Africa.

Our analyses show that patterns of haplotype sharing across the sub-Sahara can be characterised by historical gene-flow events involving groups with ancestry from across and outside of the continent. We have identified gene-flow across Africa, implying that haplotypes have been moving over (potentially large) distances in a relatively short amount of time. As a rough estimate, given that events in southern African groups involving Bantu sources have occurred within the last 2000 years (*Figure 6*) and the distance between Cameroon and south-east Africa is around 4000km, haplotypes have moved across and into different environments at a rate of roughly 2 km/year.

### Interpreting haplotype similarity as historical admixture

Analyses that model the correlations in allele frequencies (such as those performed here in the *Allele frequency differences show widespread evidence for admixture* section) provided initial evidence that the presence of Eurasian DNA across sub-Saharan Africa is the result of gene-flow back into the continent within the last 10,000 years (*Gurdasani et al., 2014*; *Pickrell et al., 2014*; *Hodgson et al., 2014a*), and that some groups have ancient (over 5 kya) shared ancestry with hunter-gather groups (*Figure 3*) (*Gurdasani et al., 2014*). Whilst the weighted admixture LD decay curves between pairs of populations used by MALDER suggests that this admixture involved particular groups, the interpretation of such events is difficult. Firstly, because our dataset includes closely related groups, it is not always possible to identify a single best matching reference, implying that sub-Saharan African groups share some ancestry with many different extant groups. On the basis of these analyses alone, it is not possible to characterise the composition of admixture sources. Secondly, when ancient events are identified with MALDER, such as in the Mossi from Burkina Faso, where we estimate admixture around 5000 years ago between a Eurasian (GBR) and a Khoesan speaking group (/Gui // Gana), we know that modern haplotypes are likely to only be an approximation of ancestral diversity (*Pickrell and Reich, 2014*). Even the Ju/'hoansi, a San group from southern Africa traditionally thought to have undergone limited recent admixture, has experienced gene-flow from non-Khoesan groups within this timeframe (*Figure 4*) (*Pickrell et al., 2012*; *2014*).

There are complications in relating admixture sources to contemporary populations. For example, our analyses indicate that the Mossi share deep ancestry with Eurasian and Khoesan groups

(*Figure 3*), but any description of the historical event leading to this observation is potentially biased by the discontinuity between extant populations and those present in Africa in the past. It is for this reason that, for older events, we define and refer to broader ancestry regions. So in this case, we describe Eurasian ancestry in general moving back into Africa, rather than British DNA in particular. GLOBETROTTER provides an alternative approach by characterising admixture as occurring between sources that themselves are mixtures of ancestry from contemporary groups. In the situation where no sample group provides a good representation of the admixture source, this additional complexity is likely to be a closer approximation to the truth, with the downside that it is not always possible to assign a specific population label to mixed admixture sources. Using contemporary populations as proxies for ancient groups is not the perfect approach and would be improved by DNA from significant numbers of ancient human individuals, at sufficient quality, with which to calibrate temporal changes in population genetics.

## Spread of genes within Africa

When new haplotypes are introduced into a population by gene-flow their fate will be partly be determined by the selective advantage they confer, as well as the chance effects of genetic drift. Selection can occur in response to a number of different factors. Greenlandic Inuit, for example, have adapted genetically to a diet rich in polyunsaturated fatty acids (*Fumagalli et al., 2015*), and one of the strongest signals of selection in the genome is found around the *LCT* gene (*Bersaglieri et al., 2004*), mutations in which allow individuals to continue to digest milk into adulthood. Responding to changes in their environment, populations living at high altitudes have adapted convergently at different genes involved in hypoxic response: at *BHLHE41* in Ethiopians (*Huerta-Sánchez et al., 2013*); *EPAS1* and *EGLN1* in Tibetans (*Yi et al., 2010*); and at a separate loci within *EGLN1* in Andean groups (*Bigham et al., 2010*). There are also several examples of humans adapting in response to infectious disease, for example at the *LARGE* gene in West Africans (*Grossman et al., 2013*), in response to pressure from Lassa fever, and at *CR1* in response to malaria (*Gurdasani et al., 2014*). Diseases such as malaria are caused by highly polymorphic parasites and movement into new environments might lead to exposure to new strains. An implication of widespread gene-flow is that it can provide a route for potentially beneficial novel mutations to enter populations allowing them to adapt to such change.

A recent example of this process is the observation of higher than expected frequencies of the Duffy-null mutation in populations from Madagascar as a result of admixture with African Bantu speaking groups (*Hodgson et al., 2014b*). The spread of the Duffy-null allele, an ancient mutation which is thought to have arose at least 30,000 years ago (*Hamblin and Di Rienzo, 2000*; *Hamblin et al., 2002*) and confers resistance to *Plasmodium vivax* malaria, throughout Africa is only possible through contact and gene-flow between populations right across the sub-Sahara. Conversely, the mutation responsible for the sickle cell phenotype, which offers protection against *P. falciparum* malaria, appears to have recently occurred five times independently in Africa, causing multiple distinct haplotypes to be observed (*Hedrick, 2011*). These mutations are young, within the order of 250–1750 years old (*Currat et al., 2002*; *Modiano et al., 2008*), so will have had limited opportunity to have been moved around by the gene-flow events that we describe. Further work is needed to understand the role of admixture in facilitating adaptation.

## Admixture and genetic epidemiology

Epidemiology is the process of identifying the mechanisms that lead to changes in disease prevalence that could result from different environments, behaviours, or genetic backgrounds. Our study helps address these questions by providing a detailed guide to genetic similarity between different ethno-liguistic groups in different geographic locations. This is equally relevant for studies of important infectious disease (such as malaria), as it is for studies of non-communicable diseases which are associated with life-style changes in developing parts of Africa (see *Rotimi and Jorde (2010)* for a review). As an example, we detect consistent genetic differences between groups in Central West Africa (e.g. the Akans and Namkam/Kasem from Ghana in *Figure 2—figure supplement 1*), but not in groups from the West and East Africa Niger-Congo ancestry regions (The Gambia and Kenya; *Figure 1—figure supplement 3*). Within these groups we see individuals with a spectrum of different ancestral backgrounds. These observations are specific to groups in our analysis, and cannot be

extended to other groups from similar populations; the seven ethnic groups from The Gambia were all collected in and around Banjul in the Western District, whereas the three ethnic groups from Ghana were collected from two hospitals in the north (Navrongo) and centre (Kumasi) of the country. Nonetheless, the potential for genetic differences to underlie difference in disease should be guided by analyses of haplotype sharing between groups.

Chromosome painting approaches also provide a quantitative measure of the extent to which self-reported ethnic labels capture genetic relationships, which is also important for controlling for the potential confounding effects of population structure (*Marchini et al., 2004*; *Price et al., 2006*), when genome-wide data are unavailable. We see that this can vary extensively from one population to the next (*Figure 1—figure supplement 3*); some individuals who report as coming from the same ethnic group cluster into different groups and other individuals from different ethnicities cluster together. We also show that there are differences in the inferred relationship between populations using analyses genotype based approaches such as PCA and $F_{ST}$, and our haplotype based analysis (*Figure 1*) and TVD (*Figure 2*). These results suggest that for some groups, haplotype similarity to other ancestries can vary more substantially than allele frequencies alone. In designing genotyping and sequencing studies these differences can be important in ensuring that the breadth of variation in African populations is adequately covered (*Kwiatkowski, 2005*; *Gurdasani et al., 2014*). Africa has an exciting and important role in furthering understanding of human biology and disease. An understanding of its patterns of genetic diversity and the historical movements of its people should help in this endeavour.

## Materials and methods

### Overview of the dataset

The dataset comprises a mixture of 2504 previously published individuals from Africa and elsewhere (see below) plus novel genotypes on 1366 sampled by the Malaria Genomic Epidemiology Network (MalariaGEN) *Figure 1—source data 1*. The MalariaGEN samples were a subset of those collected at 8 locations in Africa as part of a consortial project on genetic resistance to severe malaria: details of the study sites and investigators involved are described elsewhere (*Malaria Genomic Epidemiology Network, 2014*). Samples were genotyped on the Illumina Omni 2.5M chip in order to perform a multicentre genome-wide association study (GWAS) of severe malaria: initial GWAS findings from The Gambia, Kenya and Malawi have already been reported (*Band et al., 2013*; *Malaria Genomic Epidemiology Network, 2015*) and a manuscript describing findings at all 8 locations is in preparation.

The MalariaGEN samples used in the present analysis were selected to be representative of the main ethnic groups present at each of the 8 African study sites. We screened the samples collected at each study site (typically >1000 individuals) to select individuals whose reported parental ethnicity matched their own ethnicity. This process identified 23 ethnic groups for which we had samples for approximately 50 unrelated individuals or more. For ethnic groups with more than 50 samples available, we performed a cluster analysis on cohort-wide principle components, generated as part of the GWAS, with the R statistical programming language (*R Development Core Team, 2011*) using the *MClust* package (*Fraley et al., 2012*), choosing individuals from the cluster containing the largest number of individuals, to avoid any accidental inclusion of outlying individuals and to ensure that the 50 individuals chosen were, when possible, relatively genetically homogeneous. We note that in several ethnic groups (Malawi, the Kambe from Kenya, and the Mandinka and Fula from The Gambia; *Figure 1—figure supplement 2*) PCA of the genotype data showed a large amount of population structure. In these cases we chose two sub-groups of individuals from a given ethnic group, selected to represent the diversity of ancestry depicted by the PCs. In several other cases, following GWAS quality control (see below), genotype data for fewer than 50 control individuals were available, and in these cases we chose as many individuals as possible, regardless of the PC-based clustering or case/control status.

We additionally included further individuals from each of four Gambian ethnic groups: the Fulani, Mandinka, Jola, Wollof. The genotype data from these individuals were included as the same individuals are also being sequenced as part of The Gambia Genome Variation Project. (The Gambia Genome Variation Project will sequence a number of full genomes from four Gambian ethnic groups

as a basis for improving imputation for future West African specific GWAS.) These subsets included ~30 trios from each ethnic group, information on which was used in phasing (see below). Full genome data from these individuals will be made available in the future.

## Quality control

Detailed quality control (QC) for the MalariaGEN dataset was performed for a genome-wide association study of severe malaria in Africa and is outlined in detail elsewhere (*Malaria Genomic Epidemiology Network, 2015*). Briefly, genotype calls were formed by taking a consensus across three different calling algorithms (Illuminus, Gencall in Illumina's BeadStudio software, and GenoSNP) (*Band et al., 2013*) and were aligned to the forward strand. Using the data from each country separately, SNPs with a minor allele frequency of <1% and missingness <5% were excluded, and additional QC to account for batch effects and SNPs not in Hardy-Weinberg equilibrium was also performed.

## Combining the MalariaGEN populations with additional populations

The post-QC MalariaGEN data was combined with published data typed on the same Illumina 2.5 M Omni chip from 21 populations typed for the 1000 Genomes Project (1KGP; data downloaded on 16th October 2013 from ftp://ftp.1000genomes.ebi.ac.uk/vol1/ftp/technical/working/20120131_omni _genotypes_and_intensities/), including and accounting for duos and trios, and with publicly available data from individuals from several populations from Southern Africa (*Figure 1— source data 1*) (*1000 Genomes Project Consortium, 2012*; *Schlebusch et al., 2012*). We merged samples to the forward strand, removing any ambiguous SNPs (A to T or C to G). Merging was checked by plotting allele frequencies between populations from both datasets, which should be generally correlated (data not shown). We combined these data with further publicly available samples typed on different Illumina (Omni 1 M) chips, containing individuals from southern Africa (*Petersen et al., 2013*) and the Horn of Africa (Somalia/Ethiopia/Sudan; *Pagani et al., 2012*) to generate a final dataset containing 4216 individuals typed on 328,176 high quality common SNPs. To obtain the final set of analysis individuals we performed additional sample QC after phasing and removed American 1KGP populations (*Figure 1—source data 1*).

## Phasing

We used SHAPEITv2 (*Delaneau et al., 2012*) to generate haplotypically phased chromosomes for each individual. SHAPEITv2 conditions the underlying hidden Markov model (HMM) from *Li and Stephens (2003)* on all available haplotypes to quickly estimate haplotypic phase from genotype data. We split our dataset by chromosome and phased all individuals simultaneously, and used the most likely pairs of haplotypes (using the *–output-max* option) for each individual for downstream applications. We performed 30 iterations of the MCMC and used default values for all other parameters. As mentioned, we used known pedigree relationships to improve the phasing, using family information from both the 1KGP and The Gambia Genome Variation Project.

## Removing non-founders and cryptically related individuals

Our dataset included individuals who were known to be closely related (1KGP duos and trios; Gambia Genome Variation Project trios) and, because we took multiple samples from some population groups, there was also the potential to include cryptically related individuals. After phasing we therefore performed an additional step where we first removed all non-founders from the analysis and then identified individuals with high identity by descent (IBD), which is a measure of relatedness. Using an LD pruned set of SNPs generated by recursively removing SNPs with an $R^2 > 0.2$ using a 50 kb sliding window, we calculated the proportion of loci that are IBD for each pair of individuals in the dataset using the R package *SNPRelate* (*Zheng et al., 2012*) and estimated kinship using the pi-hat statistic (the proportion of loci that are identical for both alleles (IBD=2) plus 0.5* the proportion of loci where one allele matches (IBD=1); i.e. PI_HAT=P(IBD=2)+0.5*P(IBD=1)). For any pair of individuals where IBD > 0.2, we randomly removed one of the individuals. 327 individuals were removed during this step.

## 1KGP American populations and Native American Ancestry in 1 KG Peruvians

With the exception of Peru, post-phasing, we dropped all 1KGP American populations from the analysis dataset(97 ASW, 102 ACB, 107 CLM, 100 MXL and 111 PUR). We used a subset of the 107 Peruvian individuals that showed a large amount of putative Native American ancestry, with little apparent admixture from non-Amerindians (data not shown). Although Amerindians are not central to this study, and it is unlikely that there has been any recurrent admixture from the New World into Africa, we nevertheless generated a subset of 16 Peruvians to represent Amerindian admixture components in downstream analyses. When this subset was used, we refer to the population as PELII. The removal of these 606 American individuals left a final analysis dataset comprising 3283 individuals from 60 different population groups (*Figure 1—source data 1*).

## Analyses of population structure in sub-Saharan Africa

### Principal components analysis

We performed Principal Components Analysis (PCA) using the *SNPRelate* package in R. We removed SNPs in LD by recursively removing SNPs with an $R^2 > 0.2$, using a 50 kb sliding window, resulting in a subset of 162,322 SNPs.

### Painting chromosomes with CHROMOPAINTER

We used fineSTRUCTURE (*Lawson et al., 2012*) to identify finescale population structure and to identify high level relationships between ethnic groups. The initial step of a fineSTRUCTURE analysis involves 'painting' haplotypically phased chromosomes sequentially using an updated implementation of a model initially introduced by *Li and Stephens (2003)* and which is exploited by the CHROMOPAINTER package (*Lawson et al., 2012*). The Li and Stephens copying model explicitly relates linkage disequilibrium to the underlying recombination process and CHROMOPAINTER uses an approximate method to reconstruct each 'recipient' individual's haplotypic genome as a series of recombination 'chunks' from a set of sample 'donor' individuals. The aim of this approach is to identify, at each SNP as we move along the genome, the closest relative genome among the members of the donor sample. Because of recombination, the identity of the closest relative will change depending on the admixture history between individual genomes. Even distantly related populations share some genetic ancestry since most human genetic variation is shared (*International HapMap 3 Consortium, 2010*; *Ralph and Coop, 2013*), but the amount of shared ancestry can differ widely. We use the term 'painting' here to refer to the application of a different label to each of the donors, such that – conceptually – each donor is represented by a different colour. Donors may be coloured individually, or in groups based on *a priori* defined labels, such as the geographic population that they come from. By recovering the changing identity of the closest ancestor along chromosomes we can understand the varying contributions of different donor groups to a given population, and by understanding the distribution of these chunks we can begin to uncover the historical relationships between groups.

### Using painted chromosomes with fineSTRUCTURE

We used CHROMOPAINTER with 10 Expectation-Maximisation (E-M) steps to jointly estimate the program's parameters *Ne* and $\theta$, repeating this separately for chromosomes 1, 4, 10, and 15 and weight-averaging (using centimorgan sizes) the *Ne* and $\theta$ from the final E-M step across the four chromosomes. We performed E-M on 5 individuals from every population in the analysis and used a weighted average of the values across all pops to arrive at final values of 190.82 for Ne and 0.00045 for $\theta$. We ran each chromosome from each population separately and combined the output to generate a final coancestry matrix to be used for fineSTRUCTURE.

As the focus of our analysis is population structure within Africa, we used a 'continental force file' to combine all non-African individuals into single populations. The processing time of the algorithm is directly related to the number of individuals included in the analysis, so reducing the number of individuals speeds the analysis up. Furthermore, fineSTRUCTURE initially uses a prior that assumes that all individuals are equally distant from each other, which in the case of worldwide populations is likely to be untrue: African populations are likely to be more closely related to each other than to

non-Africa populations, for example. The result is that not all of the substructure is identified in one run.

We therefore combined all individuals from each of the non-African 1KGP populations into 'continents', which has the effect of combining all of the copying vectors from the individuals within them to look like (re-weighted) normal individuals but cannot be split and do not contribute to parameter inference, and can thus be considered as copying vectors that contain the average of the individuals within them. They are then included in the algorithm at minimal extra computational cost and exist primarily to provide chunks to (and from) the remaining groups. We combined all individuals from a labelled population (e.g. all IBS individuals were now contained in a 'continent' grouping called IBS), with the exception of the three Chinese population CHB, CHD, and CHS where we combined all individuals into a single CHN continent.

## Using fineSTRUCTURE to inform population groupings

Data was combined from various different sources, and in some groups (e.g. the Fulani) we specifically chose different groups of individuals in an attempt to cover the broad spectrum of ancestry present in that group. We used the fineSTRUCTURE tree to visually group individuals based on their ancestry. Our aim here is to try to maximise the number of individuals that we can include within an ethnic group, without merging together individuals that are distant on the tree. We also decided *not* to use the fineSTRUCTURE clusters themselves as analytical groups because of difficulties with the interpretation of the history of such clusters. We were interested in identifying the major admixture events that have occurred in the history of different populations, and it is not clear what an analytical group that is defined as, for example, a mixture of Manjago, Mandinka, and Serere individuals, would mean in our admixture analyses. In practice, this meant that we used the original geographic population labelled groups for all populations except in the Fulani and Mandinka from The Gambia, where individuals fell into two distinct groups of clusters. Here we defined two clusters for each group, with the the two groups suffixed with an 'I' or 'II' (*Figure 1—figure supplement 3*).

## Defining ancestry regions

We used a combination of genetic and ethno-linguistic information (see *Supplementary file 1* below) to define seven ancestry regions in sub-Saharan Africa. The ancestry regions are reported in *Figure 1—source data 1* and closely match the high level groupings we observed in the fineSTRUCTURE tree, with the following exceptions:

1. East African Niger-Congo speakers
   a. The two ethnic groups from Cameroon – Bantu and Semi-Bantu – were included in the Central West African Niger-Congo ancestry region despite clustering more closely with East African groups from Kenya and Tanzania in *Figure 1—figure supplement 3*. In a preliminary fineSTRUCTURE analysis based on the MalariaGEN and 1KGP populations only, using c. 1 million SNPs, the Cameroon populations clustered with other Central West African groups, and not East Africans (data not shown).
   b. Malawi was included in the South African Niger-Congo ancestry region, despite being an outlying cluster in a clade with East African Niger-Congo speaking groups. A preliminary fineSTRUCTURE analysis based on the MalariaGEN, 1KGP and Schlebusch populations, clustered Malawi with the Herero and SEBantu speakers (data not shown).
2. Southern Africa
   a. We treated Southern African individuals slightly differently: even though the fineSTRUCTURE analysis did not split them into two separate clades of Khoesan and Niger-Congo speaking individuals, we nevertheless did. *Schlebusch et al. (2012)* showed that these populations were inter-related and admixed, two properties in the data we were hoping to uncover. The final ancestry region assignments are outlined in *Figure 1—source data 1*.

## Estimating pairwise $F_{ST}$

We used *smartpca* in the EIGENSOFT (*Patterson et al., 2006*) package to estimate pairwise $F_{ST}$ between all populations. This implementation uses the Hudson estimator recently recommended by *Bhatia et al. (2013)*. Results are shown in *Figure 2*, *Figure 2—source data 2* and *Figure 2—source data 3*.

## Comparing sets of copying vectors

We used Total Variation Distance (TVD) to compare copying vectors (*Leslie et al., 2015*; *van Dorp et al., 2015*). As the copying vectors are discrete probability distributions over the same set of donors, TVD is a natural metric for quantifying the difference between them. For a given pair of groups $A$ and $B$ with copying vectors describing the copying from $n$ donors, $a$ and $b$, we can compute TVD with the following equation:

$$TVD = 0.5 \times \sum_{i=1}^{n} (|a_i - b_i|)$$

## Analyses of admixture in sub-Saharan African populations

We used a combination of approaches to explore admixture across Africa. Initially, we employed commonly used methods that utilise correlations in allele frequencies to infer historical relationships between populations. To understand ancient relationships between African groups we used the $f_3$ statistic (*Reich et al., 2009*) to look for shared drift components between a test population and two reference groups. We then used ALDER (*Loh et al., 2013*) and MALDER (*Pickrell et al., 2014*) – an updated implementation of ALDER that attempts to identify multiple admixture events – to identify admixture events through explicit modelling of admixture LD by generating weighted LD curves. The weightings of these curves are based on allele frequency differences, at varying genetic distances, between a test population and two putative admixing groups.

   To identify more recent events we used two methods which aim to more fully model the mixed ancestry in a population by utilising the distribution and length of shared tracts of ancestry as identified with the CHROMOPAINTER algorithm (*Lawson et al., 2012*; *Hellenthal et al., 2014*). We outline the details of this analysis below, but note here that, because this approach is based on the comparison and analysis of painted chromosomes, it offers a different perspective from approaches based on comparisons of allele frequencies.

### Inferring admixture with the $f_3$ statistic and ALDER

We computed the $f_3$ statistic, introduced by *Reich et al. (2009)*, as implemented in the TREEMIX package (*Pickrell and Pritchard, 2012*). These tests are a 3-population generalization of $F_{ST}$, equal to the inner product of the frequency differences between a group X and two other groups, A and B. The statistic, commonly denoted $f_3$(X:A,B) is proportional to the correlated genetic drift between A and X and A and B. If X is related in a simple way to the common ancestor with A and B, we expect this quantity to be positive. Significantly negative values of $f_3$ suggest that X has arisen as a mixture of A and B, which is thus an unambiguous signal of mixture. Standard errors are computed using a block jackknife procedure in blocks of 500 SNPs (*Supplementary file 2*).

   We used ALDER (*Patterson et al., 2012*; *Loh et al., 2013*) to test for the presence of admixture LD in different populations. This approach works by generating weighted admixture curves for pairs of populations and tests for admixture. As noted in *Loh et al. (2013)* the use of $f_3$ statistics and weighted LD curves are somewhat complementary, and there are several reasons why $f_3$ statistics might pick up signals of admixture when ALDER does not. In particular, admixture identified using $f_3$ statistics but not by ALDER is potentially related to more ancient events because whilst shared drift signals will still be present, admixture LD will have been broken over (potentially) millennia of recombination.

   As previously shown by *Loh et al. (2013)* and *Pickrell et al. (2014)*, weighted LD curves can be used to identify the source of the gene-flow by comparing curves computed using different reference populations. This is possible because theory predicts that the amplitude (i.e. the y-axis intercept) of these curves becomes larger as one uses reference populations that are closer to the true mixing populations. *Loh et al. (2013)* demonstrated that this theory holds even when using the admixed population itself as one of the reference populations. *Pickrell et al. (2014)* used this concept to identify west Eurasian ancestry in a number of East African and Khoesan speaking groups from southern Africa.

   We thus initially ran ALDER in 'one-reference' mode, where for each focal population, we generated curves involving itself with every other reference population in turn. We used the average amplitude of the curves generated in this way to identify the groups important in describing admixture in the history of the focal group. *Figure 3—figure supplement 1* shows comparative plots to

those by *Pickrell et al. (2014)* for a selection of African populations, including the Ju/'hoansi, who we also infer to have largest curve amplitudes with Eurasian groups, consistent with that previous analysis.

Next, for each focal ethnic group in turn, we used ALDER to characterise admixture using all other ethnic groups as potential reference groups (i.e. in two-population mode). In effect, this approach compares every pair of reference groups, identifying those pairs that show evidence of shared admixture LD (p<0.05 after multiple-hypothesis testing). As many of the groups are closely related, we often observed more than one pair of ethnic groups as displaying admixture in a given focal population, the results of which are highly correlated. In *Supplementary file 2*, we show the evidence for admixture only for the pair of groups with the lowest P-value for each focal group. Dates for admixture events were generated using a generation time of 29 years (*Fenner, 2005*) and the following equation:

$$D = 1950 - (n+1) * g$$

where $D$ is the inferred date of admixture, $n$ is the inferred number of generations since admixture, and $g$ is the generation time in years. To generate values comparable to the 95% date confidence intervals output by GLOBETROTTER (see below), in all plots weighted LD curve confidence intervals, which are provided as 1 standard error, were multipled by 1.96.

## Inferring multiple waves of admixture in African populations using weighted LD curves

We used MALDER (*Pickrell et al., 2014*), an implementation of ALDER designed to fit multiple exponentials to LD decay curves and therefore characterise multiple admixture events to allele frequency data. For each event we recorded (a) the curve, $C$, with the largest overall amplitude $C_{Pop1;Pop2}^{max}$, and (b) the curves which gave the largest amplitude where each of the two reference populations came from a different ancestry region, and for which a significant signal of admixture was inferred. To identify the source of an admixture event we compared curves involving populations from the same ancestry region as the two populations involved in generating $C_{Pop1;Pop2}^{max}$. For example, in the Jola, the population pair that gave $C^{max}$ were the Ju/'hoansi and GBR. Substituting these populations for their ancestry regions we get $C_{Khoesan;Eurasia}^{max}$. To understand whether this event represents a specific admixture involving the Khoesan in the history of the Jola, we identified the amplitude of the curves from (b) of the form $C_{M;Eurasia}^{max}$, where $M$ represents a population from any ancestry region other than Eurasia that gave a significant MALDER curve. We generated a Z-score for this curve comparison using the following formula (*Pickrell et al., 2014*):

$$Z = \frac{C_{Khoesan;Eurasia}^{max} - C_{Khoesan;M}^{max}}{\sqrt{se(C_{Khoesan;Eurasia}^{max})^2 + se(C_{Khoesan;M}^{max})^2}}$$

The purpose of this was to determine, for a given event, whether the sources of admixture could be represented by a single ancestry region, in which case the overall $C_{ancestry1;ancestry2}^{max}$ will be significantly greater than curves involving other regions, or whether populations from multiple ancestry regions can generate admixture curves with similar amplitudes, in which case there will be a number of ancestry regions that best represent the admixing source. We combined all values of $M$ where the Z-score computed from the above test gave a value of <2, and define the sources of admixture in this way.

To identify the major source of admixture, we performed a similar test. We determined the regional identity of the two populations used to generate $C^{max}$. In the example above, these are Khoesan and Eurasia. Separately for each region, we identify the curve, $C$, with the maximum amplitude where either of the two reference populations was from the Khoesan region, $C_{Khoesan}^{max}$ as well as the curve where neither of the reference populations was Khoesan, $C_{notKhoesan}^{max}$. We compute a Z-score as follows:

$$Z = \frac{C_{Khoesan}^{max} - C_{notKhoesan}^{max}}{\sqrt{se(C_{Khoesan}^{max})^2 + se(C_{notKhoesan}^{max})^2}}$$

This test generates two Z-scores, in this example, one for the Khoesan/not-Khoesan comparison, and one for the Eurasia/not-Eurasia comparison. We assign the main ancestry of an event to be the region(s) that generate(s) Z > 2. If neither region generates a Z-score > 2, then we do not assign a major ancestry to the event.

## Comparisons of MALDER dating using the HAPMAP worldwide and African-specific recombination maps

Recombination maps inferred from different populations are correlated on a broad scale, but differ in the fine-scale characterisation of recombination rates (*Hinch et al., 2011*). We investigated the effect of recombination map choice by recomputing MALDER results with an African specific genetic map which was inferred through patterns of LD from the HAPMAP Yoruba (YRI) sample (*Figure 3C*).

We re-inferred admixture parameters with MALDER using all populations with the African (YRI) and additionally with a European (CEU) map (*Hinch et al., 2011*). We show comparison of the dates inferred with these different maps in the main paper, and here we shows the equivalent figures to *Figure 3* for events inferred using the African (*Figure 3—figure supplement 5*) and European (*Figure 3—figure supplement 6*) maps.

## Analysis of the minimum genetic distance over which to start curve fitting when using ALDER/MALDER

A key consideration when using weighted LD to infer admixture parameters is the minimum genetic distance over which to begin computing admixture curves. Short-range LD correlations between two reference populations and a target may not only be the result of admixture, but may also be due to demography unrelated to admixture, such as shared recent bottlenecks between the target population and one of the references, or from an extended period of low population size (*Loh et al., 2013*). Indeed, the authors of the ALDER algorithm specifically incorporate checks into the default ALDER analysis pipeline that define the threshold at which a test population shares short-range LD with with either of the two reference populations. Subsequent curve analyses then ignore data from pairs of SNPs at smaller distances than this correlation threshold (*Loh et al., 2013*).

The authors nevertheless provide the option of over-riding this LD correlation threshold, allowing the user to define the minimum genetic distance over which the algorithm will begin to compute curves and therefore infer admixture. So there are (at least) two different approaches that can be used to infer admixture using weighted LD. The first is to infer the minimum distance to start building admixture curves from the data (the default), and the second is to assume that any short-range correlations that we observe in the data result from true admixture, and prescribe a minimum distance over which to infer admixture.

### Investigating correlated LD at short genetic distances

We tested these two approaches by inferring admixture using MALDER/ALDER using a minimum distance defined by the data on the one hand, and a prescribed minimum distance of 0.5cM on the other. This value is commonly used in MALDER analyses, for example by the African Genome Variation Project (*Gurdasani et al., 2014*). For each of the 48 African populations as a target, we used ALDER to infer the genetic distance over which LD correlations are shared with every other population as a reference. In *Figure 3—figure supplement 3* we show the distribution of these values across all targets for each reference population. On the basis of this analysis, to reduce the confounding effect of demography, with the exception of the next section, all ALDER/MALDER analyses presented in the paper were performed after accounting for this short range shared LD.

### Fixing a minimum genetic distance of 0.5cM with MALDER

To compare our MALDER analysis to previously published studies, we performed MALDER analyses where we fixed the minimum genetic distance to 0.5cM (*Figure 3—figure supplement 7*). The main differences between this analysis and that presented in the main part of the paper are:

1. Ancient (>5ky) admixture in Central West African populations where the main analysis found no signal of admixture
2. A second ancient admixture in Malawi c.10ky

3. More of the events appear to involve Eurasian and Khoesan groups mixing.

## Chromosome painting for mixture model and GLOBETROTTER

For the mixture model and GLOBETROTTER analyses, we generated painted samples where we disallowed closely related groups from being painting donors. In practice, this meant removing all populations from the same ancestry region as a given population from the painting analysis. The exception to this are populations from the Nilo-Saharan and Afroasiatic ancestry regions. In these groups, no population from either ancestry region was used as painting donors. We refer to this as the 'non-local' painting analysis.

## Modelling populations as mixtures of each other using linear regression

Copying vector summaries generated from painted chromosomes describe how populations relate to one another in terms of the relative time to a common shared ancestor, subsequent recent admixture, and population-specific drift (*Hellenthal et al., 2014*; *Leslie et al., 2015*). For the following analysis, we used the GLOBETROTTER package to generate the mixing coefficients used in *Figure 4*.

Given a number of potential admixing donor populations, a key step in assessing the extent of admixture in a given population $k$ is to identify which of these donors is relevant; that is, we want to identify the set $T^*$ containing all populations $l \neq k \in (1, ..., K)$ believed to be involved in any admixture generative to population $k$. Using copying vectors from the non-local painting analysis, we generate an initial estimate of the mixing coefficients that describe the copying vector of population $k$ by fitting $f^k$ as a mixture of $f^l$ where $l \neq k \in (1, ..., K)$. The purpose of this step is to assess the evidence for putative admixture in our populations, as described by *Hellenthal et al. (2014)* and *Leslie et al. (2015)*. In practice, we remove the self-copying (drift) element from these vectors, i.e. we set $f_l^k = 0$, and rescale each population's copying vector such that $\sum_{i=1}^{K} f_i^l = 1.0$ for all $l = k \in (1, ..., K)$.

We assume a standard linear model form for the relationship between $f^k$ and terms $f^l$ for $l \neq k \in (1, ..., K)$:

$$f^k = \sum_{l \neq k}^{K} \beta_l^k f^l + \epsilon$$

where $\epsilon$ is a vector of errors which we seek to choose the $\beta$ terms to minimise using non negative least squares regression with the R 'nnls' package. Here, $\beta_l^k$ is the coefficient for $f^l$ under the mixture model, and we estimate the $\beta_l^k$s under the constraints that all $\beta_l^k \geq 0$ and $\sum_{l \neq k}^{K} \beta_l^k = 1.0$. We refer to the estimated coefficient for the $l^{\text{th}}$ population as $\hat{\beta}_l^k$; to avoid over-fitting we exclude all populations for which $\hat{\beta}_l^k < 0.001$ and rescale so that $\sum_{l \neq k}^{K} \hat{\beta}_l^k = 1.0$. $T^*$ is the set of all populations whose $\hat{\beta}_l^k > 0.001$.

The $\hat{\beta}_l^k$s represent the mixing coefficients that describe a recipient population's DNA as a linear combination of the set $T^*$ donor populations. This process identifies donor populations whose copying vectors match the copying vector of the recipient, as inferred by the painting algorithm.

## Overview of GLOBETROTTER analysis pipeline

In the current setting we are interested in identifying the general historical relationships between the different African and non-African groups in our dataset. We used GLOBETROTTER (*Hellenthal et al., 2014*) to characterise patterns of ancestral gene-flow and admixture. Individuals tend to share longer stretches of DNA with more closely related individuals, so we used a focused approach where we disallowed copying from local populations.

GLOBETROTTER was originally described by *Hellenthal et al. (2014)* and a detailed description of the algorithm and the extensive validation of the method is presented in that paper. Here we run over the general framework as used in the current study, with the key difference between our approach and the default use of the algorithm being that we do not allow any groups from within the same ancestry region as a target group to be donors in the painting analysis. Throughout we use GLOBETROTTERv2.

For a given test population $k$:

1. We define the set of populations present in the same broad ancestry region as $k$ as $m$, with the caveats outlined below. Using CHROMOPAINTER, we generate painting samples using a reduced set of donors $T^*$, that included only those populations not present in the same region of Africa as $k$, i.e. $T^* = l \neq m \in (1, ..., K)$. The effect of this is to generate mosaic painted chromosomes whose ancestral chunks do not come from closely related individuals or groups, which can mask more subtle signals of admixture. For each group in turn, prior to this final painting, we ran 10 iterations of CHROMOPAINTER's EM algorithm to infer the population-specific prior copying probabilities (using the -ip flag), and use these for the final sampled paintings.

2. For each population, $l$ in $T^*$, we generate a copying-vector, $f^l$, allowing all individuals from $l$ to copy from every individual in $T^*$; i.e. we paint every population $l$ with the same set of restricted donors as $k$ in (1). For each recipient and surrogate group in turn, we sum the chunklengths donated by all individuals within all of our final donor groups (i.e. all 59 groups: including the recipient's own group) and average across all recipients to generate a single 59 element copying vector for each recipient group.

3. To account for noise due to haplotype sharing among groups, we perform a non-negative-least-squares regression (mixture model; outlined above) that takes the copying vector of the recipient group as the response and the copying vectors for each donor group as the predictors. We take the coefficients of this regression, which are restricted to be $\geq 0$ and to sum to 1 across donors, as our initial estimates of mixing coefficients describing the genetic make-up of the recipient population as a mixture of other sampled groups.

4. Within and between every pairing of 10 painting samples generated for each haploid of a recipient individual, we consider every pair of chunks (i.e. contiguous segments of DNA copied from a single donor haploid) separated by genetic distance $g$. For every two donor populations, we tabulate the number of chunk pairs where the two chunks come from the two populations. This is done in a manner to account for phasing switch errors, a common source of error when inferring haplotypes.

5. An appropriate weighting and rescaling of the curves calculated in step 4 gives us the observed coancestry curves illustrating the decay in ancestry linkage disequilibrium versus genetic distance. There is one such curve for each pair of donor populations.

6. We find the maximum likelihood estimate (MLE) of rate parameter $\lambda$ of an exponential distribution fit to all coancestry curves simultaneously. Specifically, we perform a set of linear regressions that takes each curve in turn as a response and the exponential distribution with parameter $\lambda$ as a predictor, finding the $\lambda$ that minimizes the mean-squared residuals of these regressions. This value of $\lambda$ is our estimated date of admixture. We take the coefficients from each regression. (In the case of 2 dates, we fit two independent exponential distributions with separate rate parameters to all curves simultaneously and take the MLEs of these two rate parameters as our estimates of the two respective admixture dates. We hence get two sets of coefficients, with each set representing the coefficients for one of the two exponential distributions.)

7. We perform an eigen decomposition of a matrix of values formed using the coefficients inferred in step 6. (In the case of 2 dates, we perform an eigen decomposition of each of the two matrices of coefficients, one for each inferred date.)

8. We use the eigen decomposition from step 7 and the copying vectors to infer both the proportion of admixture $\alpha$ and the mixing coefficients that describe each of the admixing source groups as a linear combination of the donor populations. (In the case of 2 dates, we perform separate fits on each of the two eigen decompositions described in step 7 to describe each admixture event separately.)

9. We re-estimate the mixing coefficients of step 3 to be $\hat{\alpha}$ times the inferred mixing coefficients of the first source plus $1 - \hat{\alpha}$ times the inferred mixing coefficients of the second source.

10. We repeat steps 5-9 for five iterations.

11. We repeat steps 4-5 using a new set of coancestry curves that should eliminate any putative signal of admixture (by taking into account the background distribution of chunks, the so-called **null procedure**), normalize our previous curves using these new ones, and repeat steps 6-10 to re-estimate dates using these normalized curves. We generate 101 date estimates via bootstrapping and assess the proportion of inferred dates that are $= 1$ or $\geq 400$, setting this proportion as our empirical $p$-value for showing *any* evidence of admixture.

12. Using values calculated in the final iteration of step 10, we classify the admixture event into one of five categories as: (A) 'no admixture', (B) 'uncertain', (C) 'one date', (D) 'multiple dates' and (E) 'one date, multiway'.

## Inferring admixture with GLOBETROTTER

We use the painting samples from (1) and the copying-vectors from (2) detailed in the pipeline above to implement GLOBETROTTER, characterising admixture in group $k$; the intuition being that any admixture observed is likely to be representative of gene-flow from across larger geographic scales.

We report the results of this analysis in *Figure 4—source data 1* as well as *Figures 4*, *5* and *6*. We generate date estimates by simultaneously fitting an exponential curve to the coancestry curves output by GLOBETROTTER and generate confidence intervals based on 100 bootstrap replicates of the GLOBETROTTER procedure, each time bootstrapping across chromosomes. Because it is unlikely that the true admixing group is present in our set of donor groups, GLOBETROTTER infers the sources of admixture as mixtures of donor groups, which are in some sense equivalent to the $\beta$ coefficients described in the mixture model approach above, but are inferred using the additional information present in the coancestry curves. We infer the composition of the admixing sources by using the $\beta$s output by GLOBETROTTER from the two (or more) sources of admixture to arrive at an understanding of the genetic basis of the the admixing source groups. These contrasts show us the contribution of each population – which we sum together into regions – to the admixture event and thus provide further intuition into historical gene-flow.

## Defining GLOBETROTTER admixture events

The GLOBETROTTER algorithm provides multiple metrics as evidence that admixture has taken place which are combined to arrive at an understanding of the nature of the observed admixture event. In particular, as the authors suggest, to generate an admixture $P$ value, we ran GLOBETROTTER's 'null' procedure, which estimates admixture parameters accounting for unusual patterns of LD, and then inferred 100 date bootstraps using this inference, identifying the proportion of inferred dates(s) that are $\leq 1$ or $\geq 400$.

Although the algorithm provides a 'best-guess' for observed admixture event, we performed the following post-GLOBETROTTER filtering to arrive at our final characterisation of events. We outline the full GLOBETROTTER output in *Figure 4—source data 2*.

1. **Southern African populations** In all Southern African groups we present the results of the GLOBETROTTER runs where results are standardised by using the null individual see *Hellenthal et al. (2014)* for further details. We also note that in both the AmaXhosa and SEBantu GLOBETROTTER found evidence for two admixture events but on running the date bootstrap inference process, in both populations the most recent date confidence interval contained 1 generation, suggesting that the dating is not reliable. Inspection of the coancestry curves in this case showed that evidence for a single date of admixture.
2. **East Africa Afroasiatic speaking populations** In all Afroasiatic groups we present the results of the GLOBETROTTER runs where results are standardised by using a 'null' individual see *Hellenthal et al. (2014)* for further details.
3. **West African Niger-Congo speaking populations** In all West African Niger-Congo speaking groups, with the exception of the Jola, GLOBETROTTER found evidence for two dates of admixture. In all cases the most recent event was young (1-10 generations) and the date bootstrap confidence interval often contained very small values. Inspection of the coancestry curves showed a sharp decrease at short genetic distances – consistent with the old inferred event – but there was little evidence of a more recent event based on these curves. In groups from this region we therefore show inference of a single date, which we take to be the older of the two dates inferred by GLOBETROTTER.

In all other cases we used the result output by GLOBETROTTER using the default approach.

## Comparing weighted LD curve dates with GLOBETROTTER dates

Noting that there were differences between the dates inferred by the two dating methods we employed, we compared the dates generated by ALDER/MALDER with those inferred from

GLOBETROTTER. *Figure 4B* shows a comparison of dates using the MALDER event inference; that is, for each population, we used the MALDER inference (either one or two dates) and used the corresponding GLOBETROTTER date inference (either one or two dates) irrespective of whether GLOBETROTTER's inferred event was different to that of MALDER. *Figure 4C* is the opposite: we use GLOBETROTTER's event inference to define whether we select one or two dates, and then use MALDER's two date inferences if two dates are inferred, or ALDER's inference if MALDER infers two dates and GLOBETROTTER infers one. Each point represents a comparison of dates for a single ethnic group, with the symbol and colour reflecting the identity of the ethnic group as in previous plots.

## Analysis of admixture using sets of restricted surrogates

Recall that for GLOBETROTTER analyses two painting steps are required. One needs to (a) paint 'recipient' individuals with a set of 'donor' individuals to generate mosaic painted chromosomes, and (b) paint all potential 'surrogate' groups with the same set of painting donors, such that we then describe admixture in the recipient individuals with this particular set of surrogate groups. One major benefit of GLOBETROTTER is its ability to represent admixing source groups as mixtures of surrogates.

## Removing non-local surrogates

In the main analysis we inferred admixture in each of the 48 target sub-Saharan African ethnic groups using all other 47 sub-Saharan African and 12 Eurasian groups as surrogates. We were interested in seeing how the admixture inference changed as we removed surrogate groups from the analysis. Masking surrogates like this provides further insight into the historical relationships between groups. By removing non-local surrogates, we can infer admixture parameters and characterize admixture sources as mixtures of this reduced set of surrogates. Given that, by definition, local groups are more closely related to the target of interest, this approach effectively asks who, outside of the targets region is best at describing the sources of admixture.

We performed several 'restricted surrogate' analyses, for different sets of targets, where we infer admixture using subsets of surrogates. One aim of the this analysis was to track the spread of Niger-Congo ancestry in the four Niger-Congo ancestry regions. For example, in the full analysis, the major sources of admixture in East African groups tended to be dominated by Southern African Niger-Congo (specifically Malawi) components. If we remove South African Niger-Congo groups from the admixture inference, how is the admixture source now composed? We performed the following restricted surrogate analyses:

1. **No local region**: for all 48 African groups, we re-ran GLOBETROTTER without allowing any surrogates from the same ancestry region.
2. **No local, east or south**: for groups from the East African and South African Niger-Congo ancestry regions, we disallowed groups from both East and South African Niger-Congo regions from being surrogates. In effect, this asks where in West/Central Africa is their Niger-Congo ancestry likely to come from.
3. **No local or west**: For West and Central West African groups, we disallowed both West and Central West African Niger-Congo groups from being admixture surrogates. In effect, this asks where in East/South African their ancestry comes from.
4. **No local or Malawi**: As previously noted, Malawi was included in the South Africa Niger-Congo ancestry region. There is some evidence, for example from the fineSTRUCTURE analysis, that Malawi is closely related to the East African groups. We therefore wanted to assess whether the inference of a large amount of South African Niger-Congo ancestry in the major sources of admixture in East African Niger-Congo groups was a function of the genetic proximity of Malawi to East Africa. We removed East African Niger-Congo and Malawi as surrogates, and re-inferred admixture parameters.

We describe the results of this analysis in the main text and *Supplementary file 3*.

## Plotting date densities

For each admixture event we split the admixture sources into their constituent components (i.e. we used the $\beta$ coefficients inferred by GLOBETROTTER) at the appropriate admixture proportions. For a given event, these components sum to 1. We multiplied these components by 100 to estimate the percentage of ancestry from a given event that originates from each donor group. We then assigned

each of the components the set of date bootstraps associated with the event. For example, in the Kauma we infer an admixture event with an admixture proportion $\alpha$ of 6% involving a minor source containing the following coefficients: Massai 0.02 Afar 0.15 GBR 0.26 GIH 0.57. We multiply each of these coefficients by $\alpha$ to obtain a final proportion that each group gives to the admixture event: Massai 0.5 Afar 1 GBR 1.5 GIH 3. We assign all the inferred date bootstraps for the Kauma to each of the populations in these proportions. In this example, GIH has twice the density of GBR. We then additionally sum components across the same ancestry region to finally arrive at the density plots in *Figures 5*.

### Gene-flow maps

We generated maps with the *rworldmaps* package in R. To generate arrows, we combined the inferred ancestral components (i.e. 1ST and 2ND EVENT SOURCES in *Figure 3*) for each population and estimated the proportion of a group's ancestry coming from each component, summed across all surrogates from a particular country. For example, if an admixture contained source contains components from both the Jola and Wollof (both from The Gambia), then these components were added together. As such, the arrows point from the country of component origin to the country of the recipient. We then plot only those arrows which relate to events pertaining to the different broad gene-flow events. For each map, we plot arrows for any event involving the following:

a. **Recent Western Bantu gene-flow**: any admixture source which has a component from either of the two Cameroon ethnic groups, Bantu and Semi-Bantu.
b. **Eastern Bantu gene-flow**: any admixture source which has a component from Kenya, Tanzania, Malawi, or South Africa (Niger-Congo speakers).
c. **East / West gene-flow**: any admixture event which has a component from Gambia, Burkina Faso, Ghana, Mali, Nigeria, Ethiopia, Sudan or Somalia.
d. **Eurasian gene-flow into Africa**: any admixture event which has a component from any Eurasian population.

An alternative map stratified by time window, rather than admixture component is shown in *Figure 6—figure supplement 1*.

### Analysis and plotting code

Code used for analyses and plotting is available at https://github.com/georgebusby/admixture_in_africa.

## Acknowledgements

We thank all the MalariaGEN study sites that contributed samples to this analysis: a list of researchers involved at each study site can be found at https://www.malariagen.net/projects/host/consortium-members.

MalariaGEN is funded by the Wellcome Trust (WT077383/Z/05/Z, 090770/Z/09/Z) and the Bill and Melinda Gates Foundation through the Foundation for the National Institutes of Health (566). Genotyping was performed at the Wellcome Trust Sanger Institute, partly funded by its core award from the Wellcome Trust (098051/Z/05/Z). This research was also supported by Centre grants from the Wellcome Trust (090532/Z/09/Z) and the Medical Research Council (G0600718). CCAS. was supported by a Wellcome Trust Career Development Fellowship (097364/Z/11/Z).

The Malaria Research and Training Center–Bandiagara Malaria Project (MRTC-BMP) in Mali group is supported by an Interagency Committee on Disability Research (ICDR) grant from the National Institute of Allergy and Infectious Diseases/US National Institutes of Health (NIAID/NIH) to the University of Maryland and the University of Bamako (USTTB) and by the Mali-NIAID/NIH International Centers for Excellence in Research (ICER) at USTTB. The Kenya Medical Research Institute (KEMRI)–Wellcome Trust Programme is funded through core support from the Wellcome Trust. This paper is published with the permission of the director of KEMRI. CMN is supported through a strategic award to the KEMRI–Wellcome Trust Programme from the Wellcome Trust (084538). The Joint Malaria Programme, Kilimanjaro Christian Medical Centre in Tanzania received funding from a UK MRC grant (G9901439).

We thank Clare Bycroft, Lucy van Dorp, and Cristian Capelli for critically evaluating the manuscript, Francesco Montinaro for insightful discussions on interpretation of MALDER analyses, and Alexander Mee-Woong Kim for suggesting (via Twitter) that the Eurasian admixture in East Africa may be explained by an Austronesian migration. The full merged and computationally phased dataset of 4216 individuals typed at 328,000 SNPs will be made available at www.malariagen.net/data. Genotypes for the MalariaGEN samples included in this paper are a subset of a large study on malaria susceptibility, the data from which will be made available at the European Nucleotide Archive (accession number TBC).

## Additional information

### Group author details

Malaria Genomic Epidemiology Network

Aaron Vanderwal: Wellcome Trust Centre for Human Genetics, University of Oxford, Oxford, United Kingdom; Wellcome Trust Sanger Institute, Hinxton, United Kingdom; Abier Elzein: Wellcome Trust Centre for Human Genetics, University of Oxford, Oxford, United Kingdom; Wellcome Trust Sanger Institute, Hinxton, United Kingdom; Aceme Nyika: Wellcome Trust Centre for Human Genetics, University of Oxford, Oxford, United Kingdom; Wellcome Trust Sanger Institute, Hinxton, United Kingdom; Alieu Mendy: Wellcome Trust Centre for Human Genetics, University of Oxford, Oxford, United Kingdom; Wellcome Trust Sanger Institute, Hinxton, United Kingdom; Alistair Miles: Wellcome Trust Centre for Human Genetics, University of Oxford, Oxford, United Kingdom; Wellcome Trust Sanger Institute, Hinxton, United Kingdom; Andrea Diss: Wellcome Trust Centre for Human Genetics, University of Oxford, Oxford, United Kingdom; Wellcome Trust Sanger Institute, Hinxton, United Kingdom; Angeliki Kerasidou: Wellcome Trust Centre for Human Genetics, University of Oxford, Oxford, United Kingdom; Wellcome Trust Sanger Institute, Hinxton, United Kingdom; Angie Green: Wellcome Trust Centre for Human Genetics, University of Oxford, Oxford, United Kingdom; Wellcome Trust Sanger Institute, Hinxton, United Kingdom; Anna E Jeffreys: Wellcome Trust Centre for Human Genetics, University of Oxford, Oxford, United Kingdom; Wellcome Trust Sanger Institute, Hinxton, United Kingdom; Bronwyn MacInnis: Wellcome Trust Centre for Human Genetics, University of Oxford, Oxford, United Kingdom; Wellcome Trust Sanger Institute, Hinxton, United Kingdom; Catherine Hughes: Wellcome Trust Centre for Human Genetics, University of Oxford, Oxford, United Kingdom; Wellcome Trust Sanger Institute, Hinxton, United Kingdom; Catherine Moyes: Wellcome Trust Centre for Human Genetics, University of Oxford, Oxford, United Kingdom; Wellcome Trust Sanger Institute, Hinxton, United Kingdom; Chris CA Spencer: Wellcome Trust Centre for Human Genetics, University of Oxford, Oxford, United Kingdom; Wellcome Trust Sanger Institute, Hinxton, United Kingdom; Christina Hubbart: Wellcome Trust Centre for Human Genetics, University of Oxford, Oxford, United Kingdom; Wellcome Trust Sanger Institute, Hinxton, United Kingdom; Cinzia Malangone: Wellcome Trust Centre for Human Genetics, University of Oxford, Oxford, United Kingdom; Wellcome Trust Sanger Institute, Hinxton, United Kingdom; Claire Potter: Wellcome Trust Centre for Human Genetics, University of Oxford, Oxford, United Kingdom; Wellcome Trust Sanger Institute, Hinxton, United Kingdom; Daniel Mead: Wellcome Trust Centre for Human Genetics, University of Oxford, Oxford, United Kingdom; Wellcome Trust Sanger Institute, Hinxton, United Kingdom; David Barnwell: Wellcome Trust Centre for Human Genetics, University of Oxford, Oxford, United Kingdom; Wellcome Trust Sanger Institute, Hinxton, United Kingdom; Dominic P Kwiatkowski: Wellcome Trust Centre for Human Genetics, University of Oxford, Oxford, United Kingdom; Wellcome Trust Sanger Institute, Hinxton, United Kingdom; Dushyanth Jyothi: Wellcome Trust Centre for Human Genetics, University of Oxford, Oxford, United Kingdom; Wellcome Trust Sanger Institute, Hinxton, United Kingdom; Eleanor Drury: Wellcome Trust Centre for Human Genetics, University of Oxford, Oxford, United Kingdom; Wellcome Trust Sanger Institute, Hinxton, United Kingdom; Elilan Somaskantharajah: Wellcome Trust Centre for Human Genetics, University of Oxford, Oxford, United Kingdom; Wellcome Trust Sanger Institute, Hinxton, United Kingdom; Eliza Hilton: Wellcome Trust Centre for Human Genetics, University of Oxford, Oxford, United Kingdom; Wellcome Trust Sanger Institute, Hinxton, United Kingdom; Ellen Leffler: Wellcome Trust Centre for Human Genetics, University of

Oxford, Oxford, United Kingdom; Wellcome Trust Sanger Institute, Hinxton, United Kingdom; Gareth Maslen: Wellcome Trust Centre for Human Genetics, University of Oxford, Oxford, United Kingdom; Wellcome Trust Sanger Institute, Hinxton, United Kingdom; Gavin Band: Wellcome Trust Centre for Human Genetics, University of Oxford, Oxford, United Kingdom; Wellcome Trust Sanger Institute, Hinxton, United Kingdom; George Busby: Wellcome Trust Centre for Human Genetics, University of Oxford, Oxford, United Kingdom; Wellcome Trust Sanger Institute, Hinxton, United Kingdom; Geraldine M Clarke: Wellcome Trust Centre for Human Genetics, University of Oxford, Oxford, United Kingdom; Wellcome Trust Sanger Institute, Hinxton, United Kingdom; Ioannis Ragoussis: Wellcome Trust Centre for Human Genetics, University of Oxford, Oxford, United Kingdom; Wellcome Trust Sanger Institute, Hinxton, United Kingdom; Jacob Almagro Garcia: Wellcome Trust Centre for Human Genetics, University of Oxford, Oxford, United Kingdom; Wellcome Trust Sanger Institute, Hinxton, United Kingdom; Jane Rogers: Wellcome Trust Centre for Human Genetics, University of Oxford, Oxford, United Kingdom; Wellcome Trust Sanger Institute, Hinxton, United Kingdom; Jantina deVries: Wellcome Trust Centre for Human Genetics, University of Oxford, Oxford, United Kingdom; Wellcome Trust Sanger Institute, Hinxton, United Kingdom; Jennifer Shelton: Wellcome Trust Centre for Human Genetics, University of Oxford, Oxford, United Kingdom; Wellcome Trust Sanger Institute, Hinxton, United Kingdom; Jiannis Ragoussis: Wellcome Trust Centre for Human Genetics, University of Oxford, Oxford, United Kingdom; Wellcome Trust Sanger Institute, Hinxton, United Kingdom; Jim Stalker: Wellcome Trust Centre for Human Genetics, University of Oxford, Oxford, United Kingdom; Wellcome Trust Sanger Institute, Hinxton, United Kingdom; Joanne Rodford: Wellcome Trust Centre for Human Genetics, University of Oxford, Oxford, United Kingdom; Wellcome Trust Sanger Institute, Hinxton, United Kingdom; John O'Brien: Wellcome Trust Centre for Human Genetics, University of Oxford, Oxford, United Kingdom; Wellcome Trust Sanger Institute, Hinxton, United Kingdom; Julie Evans: Wellcome Trust Centre for Human Genetics, University of Oxford, Oxford, United Kingdom; Wellcome Trust Sanger Institute, Hinxton, United Kingdom; Kate Rowlands: Wellcome Trust Centre for Human Genetics, University of Oxford, Oxford, United Kingdom; Wellcome Trust Sanger Institute, Hinxton, United Kingdom; Katharine Cook: Wellcome Trust Centre for Human Genetics, University of Oxford, Oxford, United Kingdom; Wellcome Trust Sanger Institute, Hinxton, United Kingdom; Kathryn Fitzpatrick: Wellcome Trust Centre for Human Genetics, University of Oxford, Oxford, United Kingdom; Wellcome Trust Sanger Institute, Hinxton, United Kingdom; Katja Kivinen: Wellcome Trust Centre for Human Genetics, University of Oxford, Oxford, United Kingdom; Wellcome Trust Sanger Institute, Hinxton, United Kingdom; Kerrin Small: Wellcome Trust Centre for Human Genetics, University of Oxford, Oxford, United Kingdom; Wellcome Trust Sanger Institute, Hinxton, United Kingdom; Kimberly J Johnson: Wellcome Trust Centre for Human Genetics, University of Oxford, Oxford, United Kingdom; Wellcome Trust Sanger Institute, Hinxton, United Kingdom; Kirk A Rockett: Wellcome Trust Centre for Human Genetics, University of Oxford, Oxford, United Kingdom; Wellcome Trust Sanger Institute, Hinxton, United Kingdom; Lee Hart: Wellcome Trust Centre for Human Genetics, University of Oxford, Oxford, United Kingdom; Wellcome Trust Sanger Institute, Hinxton, United Kingdom; Magnus Manske: Wellcome Trust Centre for Human Genetics, University of Oxford, Oxford, United Kingdom; Wellcome Trust Sanger Institute, Hinxton, United Kingdom; Marilyn McCreight: Wellcome Trust Centre for Human Genetics, University of Oxford, Oxford, United Kingdom; Wellcome Trust Sanger Institute, Hinxton, United Kingdom; Marryat Stevens: Wellcome Trust Centre for Human Genetics, University of Oxford, Oxford, United Kingdom; Wellcome Trust Sanger Institute, Hinxton, United Kingdom; Matti Pirinen: Wellcome Trust Centre for Human Genetics, University of Oxford, Oxford, United Kingdom; Wellcome Trust Sanger Institute, Hinxton, United Kingdom; Meike Hennsman: Wellcome Trust Centre for Human Genetics, University of Oxford, Oxford, United Kingdom; Wellcome Trust Sanger Institute, Hinxton, United Kingdom; Michael Parker: Wellcome Trust Centre for Human Genetics, University of Oxford, Oxford, United Kingdom; Wellcome Trust Sanger Institute, Hinxton, United Kingdom; Miguel SanJoaquin: Wellcome Trust Centre for Human Genetics, University of Oxford, Oxford, United Kingdom; Wellcome Trust Sanger Institute, Hinxton, United Kingdom; Nuno Seplúveda: Wellcome Trust Centre for Human Genetics, Wellcome Trust Sanger Institute, Hinxton, United Kingdom; Wellcome Trust Sanger Institute, ,Hinxton, ,United Kingdom; Olivia Cook: Wellcome Trust Centre for Human Genetics, University of Oxford, Oxford, United Kingdom; Wellcome Trust Sanger Institute, Hinxton, United Kingdom; Olivo Miotto: Wellcome Trust Centre for Human Genetics, University of Oxford, Oxford,

United Kingdom; Wellcome Trust Sanger Institute, Hinxton, United Kingdom; Panos Deloukas: Wellcome Trust Centre for Human Genetics, University of Oxford, Oxford, United Kingdom; Wellcome Trust Sanger Institute, Hinxton, United Kingdom; Rachel Craik: Wellcome Trust Centre for Human Genetics, University of Oxford, Oxford, United Kingdom; Wellcome Trust Sanger Institute, Hinxton, United Kingdom; Rebecca Wrigley: Wellcome Trust Centre for Human Genetics, University of Oxford, Oxford, United Kingdom; Wellcome Trust Sanger Institute, Hinxton, United Kingdom; Renee Watson: Wellcome Trust Centre for Human Genetics, University of Oxford, Oxford, United Kingdom; Wellcome Trust Sanger Institute, Hinxton, United Kingdom; Richard Pearson: Wellcome Trust Centre for Human Genetics, University of Oxford, Oxford, United Kingdom; Wellcome Trust Sanger Institute, Hinxton, United Kingdom; Robert Hutton: Wellcome Trust Centre for Human Genetics, University of Oxford, Oxford, United Kingdom; Wellcome Trust Sanger Institute, Hinxton, United Kingdom; Samuel Oyola: Wellcome Trust Centre for Human Genetics, University of Oxford, Oxford, United Kingdom; Wellcome Trust Sanger Institute, Hinxton, United Kingdom; Sarah Auburn: Wellcome Trust Centre for Human Genetics, University of Oxford, Oxford, United Kingdom; Wellcome Trust Sanger Institute, Hinxton, United Kingdom; Shivang Shah: Wellcome Trust Centre for Human Genetics, University of Oxford, Oxford, United Kingdom; Wellcome Trust Sanger Institute, Hinxton, United Kingdom; Si Quang Le: Wellcome Trust Centre for Human Genetics, University of Oxford, Oxford, United Kingdom; Wellcome Trust Sanger Institute, Hinxton, United Kingdom; Sile Molloy: Wellcome Trust Centre for Human Genetics, University of Oxford, Oxford, United Kingdom; Wellcome Trust Sanger Institute, Hinxton, United Kingdom; Susan Bull: Wellcome Trust Centre for Human Genetics, University of Oxford, Oxford, United Kingdom; Wellcome Trust Sanger Institute, Hinxton, United Kingdom; Susana Campino: Wellcome Trust Centre for Human Genetics, University of Oxford, Oxford, United Kingdom; Wellcome Trust Sanger Institute, Hinxton, United Kingdom; Taane G Clark: Wellcome Trust Centre for Human Genetics, University of Oxford, Oxford, United Kingdom; Wellcome Trust Sanger Institute, Hinxton, United Kingdom; Valentín Ruano-Rubio: Wellcome Trust Centre for Human Genetics, Wellcome Trust Sanger Institute, Hinxton, United Kingdom; Wellcome Trust Sanger Institute, ,Hinxton, ,United Kingdom; Victoria Cornelius: Wellcome Trust Centre for Human Genetics, University of Oxford, Oxford, United Kingdom; Wellcome Trust Sanger Institute, Hinxton, United Kingdom; Yik Ying Teo: Wellcome Trust Centre for Human Genetics, University of Oxford, Oxford, United Kingdom; Wellcome Trust Sanger Institute, Hinxton, United Kingdom; Patrick Corran: National Institute for Biological Standards, South Mimms, United Kingdom; Nilupa De Silva: National Institute for Biological Standards, South Mimms, United Kingdom; Paul Risley: National Institute for Biological Standards, South Mimms, United Kingdom; Alan Doyle: The Wellcome Trust, London, United Kingdom; Jennifer Evans: Bernhard-Nocht-Institut für Tropenmedizin, Hamburg, Germany; Rolf Horstmann: Bernhard-Nocht-Institut für Tropenmedizin, Hamburg, Germany; Chris Plowe: Howard Hughes Medical Institute, University of Maryland, Baltimore, United States; Patrick Duffy: National Institute of Allergy and Infectious Diseases, Bethesda, United States; Dan Carucci: Foundation for the National Institutes of Health, Bethesda, United States; Michael Gottleib: Foundation for the National Institutes of Health, Bethesda, United States; Adama Tall: The Institut Pasteur de Dakar, Dakar, Senegal; The Institut Pasteur de Dakar, Paris, France; Alioune Badara Ly: The Institut Pasteur de Dakar, Dakar, Senegal; The Institut Pasteur de Dakar, Paris, France; Amagana Dolo: The Institut Pasteur de Dakar, Dakar, Senegal; The Institut Pasteur de Dakar, Paris, France; Anavaj Sakuntabhai: The Institut Pasteur de Dakar, Dakar, Senegal; The Institut Pasteur de Dakar, Paris, France; Odile Puijalon: The Institut Pasteur de Dakar, Dakar, Senegal; The Institut Pasteur de Dakar, Paris, France; Abdou Bah: Medical Research Council Laboratories, Banjul, Gambia; Abdoulie Camara: Medical Research Council Laboratories, Banjul, Gambia; Abubacar Sadiq: Medical Research Council Laboratories, Banjul, Gambia; Aja Abie Khan: Medical Research Council Laboratories, Banjul, Gambia; Amie Jobarteh: Medical Research Council Laboratories, Banjul, Gambia; Anthony Mendy: Medical Research Council Laboratories, Banjul, Gambia; Augustine Ebonyi: Medical Research Council Laboratories, Banjul, Gambia; Bakary Danso: Medical Research Council Laboratories, Banjul, Gambia; Bintou Taal: Medical Research Council Laboratories, Banjul, Gambia; Climent Casals-Pascual: Medical Research Council Laboratories, Banjul, Gambia; David J Conway: Medical Research Council Laboratories, Banjul, Gambia; Emmanuel Onykwelu: Medical Research Council Laboratories, Banjul, Gambia; Fatoumatta Sisay-Joof: Medical Research Council Laboratories, Banjul, Gambia; Giorgio Sirugo: Medical Research Council Laboratories, Banjul, Gambia; Haddy Kanyi: Medical Research Council Laboratories, Banjul,

Gambia; Haddy Njie: Medical Research Council Laboratories, Banjul, Gambia; Herbert Obu: Medical Research Council Laboratories, Banjul, Gambia; Horeja Saine: Medical Research Council Laboratories, Banjul, Gambia; Idrissa Sambou: Medical Research Council Laboratories, Banjul, Gambia; Ismaela Abubakar: Medical Research Council Laboratories, Banjul, Gambia; Jalimory Njie: Medical Research Council Laboratories, Banjul, Gambia; Janet Fullah: Medical Research Council Laboratories, Banjul, Gambia; Jula Jaiteh: Medical Research Council Laboratories, Banjul, Gambia; Kalifa A Bojang: Medical Research Council Laboratories, Banjul, Gambia; Kebba Jammeh: Medical Research Council Laboratories, Banjul, Gambia; Kumba Sabally-Ceesay: Medical Research Council Laboratories, Banjul, Gambia; Lamin Manneh: Medical Research Council Laboratories, Banjul, Gambia; Landing Camara: Medical Research Council Laboratories, Banjul, Gambia; Lawrence Yamoah: Medical Research Council Laboratories, Banjul, Gambia; Madi Njie: Medical Research Council Laboratories, Banjul, Gambia; Malick Njie: Medical Research Council Laboratories, Banjul, Gambia; Margaret Pinder: Medical Research Council Laboratories, Banjul, Gambia; Mariatou Jallow: Medical Research Council Laboratories, Banjul, Gambia; Mohammed Aiyegbo: Medical Research Council Laboratories, Banjul, Gambia; Momodou Jasseh: Medical Research Council Laboratories, Banjul, Gambia; Momodou Lamin Keita: Medical Research Council Laboratories, Banjul, Gambia; Momodou Saidy-Khan: Medical Research Council Laboratories, Banjul, Gambia; Muminatou Jallow: Medical Research Council Laboratories, Banjul, Gambia; Ndey Ceesay: Medical Research Council Laboratories, Banjul, Gambia; Oba Rasheed: Medical Research Council Laboratories, Banjul, Gambia; Pa Lamin Ceesay: Medical Research Council Laboratories, Banjul, Gambia; Pamela Esangbedo: Medical Research Council Laboratories, Banjul, Gambia; Ramou Cole-Ceesay: Medical Research Council Laboratories, Banjul, Gambia; Rasaq Olaosebikan: Medical Research Council Laboratories, Banjul, Gambia; Simon Correa: Medical Research Council Laboratories, Banjul, Gambia; Sophie Njie: Medical Research Council Laboratories, Banjul, Gambia; Stanley Usen: Medical Research Council Laboratories, Banjul, Gambia; Yaya Dibba: Medical Research Council Laboratories, Banjul, Gambia; Abdoulaye Barry: University Of Bamako, Bamako, Mali; Abdoulaye Djimdé: University Of Bamako, Bamako, Mali; Abdourahmane H Sall: University Of Bamako, Bamako, Mali; Amadou Abathina: University Of Bamako, Bamako, Mali; Amadou Niangaly: University Of Bamako, Bamako, Mali; Awa Dembele: University Of Bamako, Bamako, Mali; Belco Poudiougou: University Of Bamako, Bamako, Mali; Elizabeth Diarra: University Of Bamako, Bamako, Mali; Kariatou Bamba: University Of Bamako, Bamako, Mali; Mahamadou A Thera: University Of Bamako, Bamako, Mali; Ogobara Doumbo: University Of Bamako, Bamako, Mali; Ousmane Toure: University Of Bamako, Bamako, Mali; Salimata Konate: University Of Bamako, Bamako, Mali; Sibiry Sissoko: University Of Bamako, Bamako, Mali; Mahamadou Diakite: University Of Bamako, Bamako, Mali; Amadou T Konate: Centre National De Recherche Et De Formation Sur Le Paludisme, Ouagadougou, Burkina Faso; David Modiano: Centre National De Recherche Et De Formation Sur Le Paludisme, Ouagadougou, Burkina Faso; Edith C Bougouma: Centre National De Recherche Et De Formation Sur Le Paludisme, Ouagadougou, Burkina Faso; Germana Bancone: Centre National De Recherche Et De Formation Sur Le Paludisme, Ouagadougou, Burkina Faso; Issa N Ouedraogo: Centre National De Recherche Et De Formation Sur Le Paludisme, Ouagadougou, Burkina Faso; Jaques Simpore: Centre National De Recherche Et De Formation Sur Le Paludisme, Ouagadougou, Burkina Faso; Sodiomon B Sirima: Centre National De Recherche Et De Formation Sur Le Paludisme, Ouagadougou, Burkina Faso; Valentina D Mangano: Centre National De Recherche Et De Formation Sur Le Paludisme, Ouagadougou, Burkina Faso; Marita Troye-Blomberg: Centre National De Recherche Et De Formation Sur Le Paludisme, Ouagadougou, Burkina Faso; Abraham R Oduro: Noguchi Memorial Institute For Medical Research, Legon, Ghana; Navrongo Health Research Centre, Navrongo, Ghana; Abraham V O Hodgson: Noguchi Memorial Institute For Medical Research, Legon, Ghana; Navrongo Health Research Centre, Navrongo, Ghana; Anita Ghansah: Noguchi Memorial Institute For Medical Research, Legon, Ghana; Navrongo Health Research Centre, Navrongo, Ghana; Francis Nkrumah: Noguchi Memorial Institute For Medical Research, Legon, Ghana; Navrongo Health Research Centre, Navrongo, Ghana; Frank Atuguba: Noguchi Memorial Institute For Medical Research, Legon, Ghana; Navrongo Health Research Centre, Navrongo, Ghana; Kwadwo A Koram: Noguchi Memorial Institute For Medical Research, Legon, Ghana; Navrongo Health Research Centre, Navrongo, Ghana; Lucas N Amenga-Etego: Noguchi Memorial Institute For Medical Research, Legon, Ghana; Navrongo Health Research Centre, Navrongo, Ghana; Michael D Wilson: Noguchi Memorial Institute For Medical Research, Legon,

Ghana; Navrongo Health Research Centre, Navrongo, Ghana; Nana Akosua Ansah: Noguchi Memorial Institute For Medical Research, Legon, Ghana; Navrongo Health Research Centre, Navrongo, Ghana; Nathan Mensah: Noguchi Memorial Institute For Medical Research, Legon, Ghana; Navrongo Health Research Centre, Navrongo, Ghana; Patrick A Ansah: Noguchi Memorial Institute For Medical Research, Legon, Ghana; Navrongo Health Research Centre, Navrongo, Ghana; Thomas Anyorigiya: Noguchi Memorial Institute For Medical Research, Legon, Ghana; Navrongo Health Research Centre, Navrongo, Ghana; Victor Asoala: Noguchi Memorial Institute For Medical Research, Legon, Ghana; Navrongo Health Research Centre, Navrongo, Ghana; William O Rogers: Noguchi Memorial Institute For Medical Research, Legon, Ghana; Navrongo Health Research Centre, Navrongo, Ghana; Alex Osei Akoto: Kwame Nkrumah University Of Science And Technology, Kumasi, Ghana; Alex Owusu Ofori: Kwame Nkrumah University Of Science And Technology, Kumasi, Ghana; Anthony Enimil: Kwame Nkrumah University Of Science And Technology, Kumasi, Ghana; Daniel Ansong: Kwame Nkrumah University Of Science And Technology, Kumasi, Ghana; David Sambian: Kwame Nkrumah University Of Science And Technology, Kumasi, Ghana; Emmanuel Asafo-Agyei: Kwame Nkrumah University Of Science And Technology, Kumasi, Ghana; Justice Sylverken: Kwame Nkrumah University Of Science And Technology, Kumasi, Ghana; Sampson Antwi: Kwame Nkrumah University Of Science And Technology, Kumasi, Ghana; Tsiri Agbenyega: Kwame Nkrumah University Of Science And Technology, Kumasi, Ghana; Adebola E Orimadegun: University Of Ibadan, Ibadan, Nigeria; Folakemi Anjol Amodu: University Of Ibadan, Ibadan, Nigeria; Olajumoke Oni: University Of Ibadan, Ibadan, Nigeria; Olayemi O Omotade: University Of Ibadan, Ibadan, Nigeria; Olukemi Amodu: University Of Ibadan, Ibadan, Nigeria; Subulade Olaniyan: University Of Ibadan, Ibadan, Nigeria; Andre Ndi: University Of Buea, Buea, Cameroon; Clarisse Yafi: University Of Buea, Buea, Cameroon; Eric Akum Achidi: University Of Buea, Buea, Cameroon; Eric Mbunwe: University Of Buea, Buea, Cameroon; Judith Anchang-Kimbi: University Of Buea, Buea, Cameroon; Regina Mugri: University Of Buea, Buea, Cameroon; Richard Besingi: University Of Buea, Buea, Cameroon; Tobias O Apinjoh: University Of Buea, Buea, Cameroon; Vincent Titanji: University Of Buea, Buea, Cameroon; Ahmed Elhassan: University Of Khatoum, Khartoum, Sudan; Ayman Hussein: University Of Khatoum, Khartoum, Sudan; Hiba Mohamed: University Of Khatoum, Khartoum, Sudan; Ibrahim Elhassan: University Of Khatoum, Khartoum, Sudan; Muntaser Ibrahim: University Of Khatoum, Khartoum, Sudan; Gilbert Kokwaro: KEMRI-Wellcome Research Programme, Kilifi, Kenya; Tom Oluoch: KEMRI-Wellcome Research Programme, Kilifi, Kenya; Alexander Macharia: KEMRI-Wellcome Research Programme, Kilifi, Kenya; Carolyne M Ndila: KEMRI-Wellcome Research Programme, Kilifi, Kenya; Charles Newton: KEMRI-Wellcome Research Programme, Kilifi, Kenya; Daniel H Opi: KEMRI-Wellcome Research Programme, Kilifi, Kenya; Dorcas Kamuya: KEMRI-Wellcome Research Programme, Kilifi, Kenya; Evasius Bauni: KEMRI-Wellcome Research Programme, Kilifi, Kenya; Kevin Marsh: KEMRI-Wellcome Research Programme, Kilifi, Kenya; Norbert Peshu: KEMRI-Wellcome Research Programme, Kilifi, Kenya; Sassy Molyneux: KEMRI-Wellcome Research Programme, Kilifi, Kenya; Sophie Uyoga: KEMRI-Wellcome Research Programme, Kilifi, Kenya; Thomas N Williams: KEMRI-Wellcome Research Programme, Kilifi, Kenya; Vicki Marsh: KEMRI-Wellcome Research Programme, Kilifi, Kenya; Alphaxard Manjurano: Joint Malaria Programme, Kilimanjaro Christian Medical Centre, London, United Kingdom; London School Of Hygiene And Tropical Medicine, Moshi, Tanzania; Behzad Nadjm: Joint Malaria Programme, Kilimanjaro Christian Medical Centre, London, United Kingdom; London School Of Hygiene And Tropical Medicine, Moshi, Tanzania; Caroline Maxwell: Joint Malaria Programme, Kilimanjaro Christian Medical Centre, London, United Kingdom; London School Of Hygiene And Tropical Medicine, Moshi, Tanzania; Chris Drakeley: Joint Malaria Programme, Kilimanjaro Christian Medical Centre, London, United Kingdom; London School Of Hygiene And Tropical Medicine, Moshi, Tanzania; Eleanor Riley: Joint Malaria Programme, Kilimanjaro Christian Medical Centre, London, United Kingdom; London School Of Hygiene And Tropical Medicine, Moshi, Tanzania; Frank Mtei: Joint Malaria Programme, Kilimanjaro Christian Medical Centre, London, United Kingdom; London School Of Hygiene And Tropical Medicine, Moshi, Tanzania; George Mtove: Joint Malaria Programme, Kilimanjaro Christian Medical Centre, London, United Kingdom; London School Of Hygiene And Tropical Medicine, Moshi, Tanzania; Hannah Wangai: Joint Malaria Programme, Kilimanjaro Christian Medical Centre, London, United Kingdom; London School Of Hygiene And Tropical Medicine, Moshi, Tanzania; Hugh Reyburn: Joint Malaria Programme, Kilimanjaro Christian Medical Centre, London, United Kingdom; London School

Of Hygiene And Tropical Medicine, Moshi, Tanzania; Sarah Joseph: Joint Malaria Programme, Kilimanjaro Christian Medical Centre, London, United Kingdom; London School Of Hygiene And Tropical Medicine, Moshi, Tanzania; Deus Ishengoma: National Institute For Medical Research, Tanga, Tanzania; Martha Lemnge: National Institute For Medical Research, Tanga, Tanzania; Theonest Mutabingwa: National Institute For Medical Research, Tanga, Tanzania; Julie Makani: Muhimbili University Of Health And Allied Sciences, The University Of Dar Es Salaam, Dar Es Salaam, Tanzania; Sharon Cox: Muhimbili University Of Health And Allied Sciences, The University Of Dar Es Salaam, Dar Es Salaam, Tanzania; Ajib Phiri: Blantyre Malaria Project With Malawi-Liverpool-Wellcome Programme, Blantyre, Malawi; Annie Munthali: Blantyre Malaria Project With Malawi-Liverpool-Wellcome Programme, Blantyre, Malawi; David Kachala: Blantyre Malaria Project With Malawi-Liverpool-Wellcome Programme, Blantyre, Malawi; Labes Njiragoma: Blantyre Malaria Project With Malawi-Liverpool-Wellcome Programme, Blantyre, Malawi; Malcolm E Molyneux: Blantyre Malaria Project With Malawi-Liverpool-Wellcome Programme, Blantyre, Malawi; Mike Moore: Blantyre Malaria Project With Malawi-Liverpool-Wellcome Programme, Blantyre, Malawi; Neema Ntunthama: Blantyre Malaria Project With Malawi-Liverpool-Wellcome Programme, Blantyre, Malawi; Paul Pensulo: Blantyre Malaria Project With Malawi-Liverpool-Wellcome Programme, Blantyre, Malawi; Terrie Taylor: Blantyre Malaria Project With Malawi-Liverpool-Wellcome Programme, Blantyre, Malawi; Vysaul Nyirongo: Blantyre Malaria Project With Malawi-Liverpool-Wellcome Programme, Blantyre, Malawi; Richard Carter: The University of Colombo, Colombo, Sri Lanka; Deepika Fernando: The University of Colombo, Colombo, Sri Lanka; Nadira Karunaweera: The University of Colombo, Colombo, Sri Lanka; Rajika Dewasurendra: The University of Colombo, Colombo, Sri Lanka; Prapat Suriyaphol: Mahidol University, Bangkok, Thailand; Pratap Singhasivanon: Mahidol University, Bangkok, Thailand; Cameron P Simmons: Oxford University Clinical Research Unit, Ho Chi Minh City, Vietnam; Cao Quang Thai: Oxford University Clinical Research Unit, Ho Chi Minh City, Vietnam; Dinh Xuan Sinh: Oxford University Clinical Research Unit, Ho Chi Minh City, Vietnam; Jeremy Farrar: Oxford University Clinical Research Unit, Ho Chi Minh City, Vietnam; Ly Van Chuong: Oxford University Clinical Research Unit, Ho Chi Minh City, Vietnam; Nguyen Hoan Phu: Oxford University Clinical Research Unit, Ho Chi Minh City, Vietnam; Nguyen T Hieu: Oxford University Clinical Research Unit, Ho Chi Minh City, Vietnam; Nguyen Thi Hoang Mai: Oxford University Clinical Research Unit, Ho Chi Minh City, Vietnam; Nguyen Thi Ngoc Quyen: Oxford University Clinical Research Unit, Ho Chi Minh City, Vietnam; Nicholas Day: Oxford University Clinical Research Unit, Ho Chi Minh City, Vietnam; Sarah J Dunstan: Oxford University Clinical Research Unit, Ho Chi Minh City, Vietnam; Sean E O'Riordan: Oxford University Clinical Research Unit, Ho Chi Minh City, Vietnam; Tran Thi Hong Chau: Oxford University Clinical Research Unit, Ho Chi Minh City, Vietnam; Tran Tinh Hien: Oxford University Clinical Research Unit, Ho Chi Minh City, Vietnam; Angela Allen: Papua New Guinea Institute For Medical Research, Madang, Papua New Guinea; Enmoore Lin: Papua New Guinea Institute For Medical Research, Madang, Papua New Guinea; Harin Karunajeewa: Papua New Guinea Institute For Medical Research, Madang, Papua New Guinea; Ivo Mueller: Papua New Guinea Institute For Medical Research, Madang, Papua New Guinea; John Reeder: Papua New Guinea Institute For Medical Research, Madang, Papua New Guinea; Laurens Manning: Papua New Guinea Institute For Medical Research, Madang, Papua New Guinea; Moses Laman: Papua New Guinea Institute For Medical Research, Madang, Papua New Guinea; Pascal Michon: Papua New Guinea Institute For Medical Research, Madang, Papua New Guinea; Peter Siba: Papua New Guinea Institute For Medical Research, Madang, Papua New Guinea; Stephen Allen: Papua New Guinea Institute For Medical Research, Madang, Papua New Guinea; Timothy M E Davis: Papua New Guinea Institute For Medical Research, Madang, Papua New Guinea

## Funding

| Funder | Grant reference number | Author |
| --- | --- | --- |
| Wellcome Trust | 084538 | Carolyne M Ndila |
| National Institute of Allergy and Infectious Diseases | | Ogobara Doumba |
| Wellcome Trust | 090532 | Dominic P Kwiatkowski |
| Medical Research Council | G0600718 | Dominic P Kwiatkowski |

| | | |
|---|---|---|
| Foundation for the National Institutes of Health | 566 | Dominic P Kwiatkowski |
| Wellcome Trust | 098051 | Dominic P Kwiatkowski |
| Wellcome Trust | 077383 | Dominic P Kwiatkowski |
| Wellcome Trust | 097364 | Chris C.A. Spencer |
| Wellcome Trust | 090770 | Dominic P Kwiatkowski |
| Medical Research Council | MR/M006212/1 | Dominic P Kwiatkowski |

The funders had no role in study design, data collection and interpretation, or the decision to submit the work for publication.

### Author contributions
GBJB, CCAS, Conception and design, Analysis and interpretation of data, Drafting or revising the article; GB, QSL, Analysis and interpretation of data, Contributed unpublished essential data or reagents; MJ, EB, VDM, LNA-E, AE, TA, CMN, AM, VN, OD, Acquisition of data, Contributed unpublished essential data or reagents; KAR, Acquisition of data, Analysis and interpretation of data, Contributed unpublished essential data or reagents; DPK, Conception and design, Acquisition of data, Analysis and interpretation of data, Drafting or revising the article

### Author ORCIDs
George BJ Busby, http://orcid.org/0000-0003-4148-6222
Kirk A Rockett, http://orcid.org/0000-0002-6369-9299

### Ethics
Human subjects: Investigators from study sites worked together to agree on principles for sharing data and standardizing clinical definitions, and to define best ethical practices across different local settings including the development of guidelines for informed consent, as described elsewhere (Malaria Genomic Epidemiology Network, Nature 2015). Further information on policies, research and the consent process may be found on the MalariaGEN website (http://www.malariagen.net/community/ethics-governance).

## Additional files

### Supplementary files
• Supplementary file 1. A note on ethnolinguistic groupings.
• Supplementary file 2. $f_3$ and ALDER analysis.
• Supplementary file 3. Summary of inferred gene-flow from West to East and South Africa.

### Major datasets
The following dataset was generated:

| Author(s) | Year | Dataset title | Dataset URL | Database, license, and accessibility information |
|---|---|---|---|---|
| Malaria Genomic Epidemiology Network | 2016 | African population structure dataset | https://www.malariagen.net/resource/18 | Publicly available at MalariaGen Genomic Epidemiology Network (www.malariagen.net) |

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
