## [Decision Letter]

Thank you for submitting your article "Admixture into and within sub-Saharan Africa" for consideration by *eLife*. Your article has been reviewed by two peer reviewers, and the evaluation has been overseen by a Reviewing Editor and Detlef Weigel as the Senior Editor.

The reviewers have discussed the reviews with one another and the Reviewing Editor has drafted this decision to help you prepare a revised submission.

Summary:

Busby et al. present new genomic data from several African populations that greatly expand our understanding of African genomic diversity.

Essential revisions:

The reviewers raised a number of important points, specifically regarding the interpretation of the analysis. There should be some additional attention paid to these points at the relevant spots in the manuscript. I (the editor) have lightly edited some of the major points raised by the reviewers below.

1) Reviewer #2 argued that the results of the analysis are highly dependent upon the available samples. While the sampling here is better than previously available, relative to the actual scale of Africa it remains under sampled in several key regions. This fact is acknowledged at several points in the manuscript, however, much of the interpretation of the results are done without considering how sampling may influence their findings. For example, a central focus of the manuscript is on the Bantu expansion, however there are no samples included from the purported area of the Western branch of their spread south. This is a major limitation that needs to be considered at all points.

2) Reviewer #2 argued that the authors' reading of the available literature on African prehistory is a bit thin at times. These analyses will be much more valuable and have a much wider impact if they are better integrated into the reasonably well developed archaeological and linguistic understanding of African population history

The primary disconnect from the interpretations presented here and the existing literature is with respect to the Bantu expansion. The authors consider their Cameroon samples to best represent the western branch of the Bantu migration. However, the archaeological and linguistic consensus seems to be that the area of Cameroon is actually the origin of the Bantu expansion. In other words, the Cameroon samples may best represent something like the ancestral population of both the Eastern and Western branches of the expansion. Also, since the Cameroonian samples are the most southern and western samples included from the West African set, it means that, contrary to the author's assertions, no western branch Bantu populations were included in this study. Considering the Cameroonian samples as representative of the origin, rather than the western part of the expansion will change the interpretation considerably.

3) MALDER seems to give consistently older dates than GLOBETROTTER? Why? Which should we prefer? The only way to properly address this question is to attempt to relate the inferred dates to the known historical or archaeological record (e.g. dates that are independent of population genetic inference). Some contextualisation here would be extremely helpful. Otherwise there is no clear way for the reader to evaluate these findings.

4) In the second paragraph of the subsection “Population movements within Africa and the Bantu expansion”. Tishkoff et al. 2009 showed pretty strongly convincingly using genomic STR markers that the Bantu expansion involved a spread of West African people, rather than simple cultural diffusion. Thus, the characterisation of this as 1) an ongoing controversy, and 2) one only addressed using uniparental markers is not correct. In fact, Tishkoff et al. did so with even greater population sampling than found in the present study.

5) In the second paragraph of the subsection “Population movements within Africa and the Bantu expansion”. The descriptions of the two linguistic hypotheses are particularly lacking in detail. It is important to put dates on the possible early and late splits, since the magnitude of the difference between these will determine the power to detect a difference genetically. Also, it is stated that recent genetic and linguistic analyses support the late split, but no mention of how. This is important for contextualising how this analysis adds to these other findings.

6) In the third paragraph of the subsection “Population movements within Africa and the Bantu expansion”. I do not understand the logic behind the claim that the early split hypothesis predicts all of the ancestry of Eastern NC speakers comes from Central West Africa. The general consensus is that the Bantu expansion originated in the area of Cameroon or Nigeria ~5kya, then split into two major forks (Greenberg 1972, Phillipson 1975). The first spread around the Congo forest to the North, and entered East Africa in the Great Lakes region and continued down the eastern margin of Africa to Southern Africa. The second went around the Congo Forest down the Western Coast and entered Southern Africa. These two branches are thought to have ultimately reunited in Southern Africa. The analysis presented here finding Southern African NC ancestry in East African NC speakers might best be explained by the former descending from the latter. The ability to detect ancestry from western branch of the Bantu expansion is not possible here because there are no populations sampled from the area of Gabon, Angola, etc. I am worried these findings are a sampling artefact.

7) In the subsection “(3) Medieval contact between Asia and the East African Swahili Coast”. Again it is not clear that this represents the Western Bantu expansion since the samples come from Cameroon which is hypothesised to be near the homeland of both the Eastern and Western branches of the expansion.

8) In the subsection “(7) Pre-Bantu pastoralist movements from East to South Africa”. The proposed model here is incongruent with the known archaeology, which shows the Bantu expansion reaching the eastern side of the Congo Forest in the north near Lake Victoria, rather than the south. The archaeology may not be correct, however given the lack of western Bantu samples included in this study, the conservative interpretation is that these findings are a sampling artefact.

9) In the first paragraph of the Discussion. The adoption of pastoralism began well before 2,000 years ago. The earliest archaeological evidence of cattle keeping may date to as early as 9kya in the Nile River Valley, and is well established by 6kya (see Boivin et al. 2010). Pastoralists reach the Koobi Fora region of Kenya by 4.5 kya, and are east of Lake Victoria by 3.8 kyat (see Pendergast 2011).

10) In the first paragraph of the Discussion. Why is the GLOBETROTTER analysis more precise? Is there any evidence to support the assertion? Also, even if more precise, is it more accurate?

11) Methods section: The Methods section is often difficult to follow because it frequently mixes results with the methods themselves. If results of one analysis are necessary to inform how the following analysis is to be performed, it would be preferable to lay out the strict methods and the logic of how methodological options were chosen in the Methods section, and then keep the actual progression of results in the Results section.

12) More generally, the manuscript is a bit too long; in particular, the Discussion and Abstract contain a substantial amount of content on genetic epidemiology and infectious disease that is only tangentially related to the analyses performed in the manuscript.

13) The description of the TVD analyses (subsection “Haplotypes reveal subtle population structure”) could be clarified. The comparison of TVD vs. Fst suggests that TVD is more informative ("expected to capture more variability") because recombination rates are faster than mutation rates. However, aren't recombination rates and mutation rates quite similar (both roughly 10^-8 per base per generation)? Also, the difference between TVD and Fst seems to be more than just amount of information captured; the two metrics must be capturing different types of information given the empirical differences between the values of the metrics. More intuition would help. Overall, I found the discussion of TVD rather confusing, especially given that paragraph ends with a caveat about interpretation – what then should we make of the TVD results in Figure 2?

14) In the main GLOBETROTTER analyses, the authors note that the GLOBETROTTER approach allows them to infer whether Eurasian haplotypes came directly into sub-Saharan Africa or were brought indirectly together with sub-Saharan groups (subsection “Direct and indirect gene flow from Eurasia back into Africa”, second paragraph). Is this statement technically correct? It seems to me that inference of this sort would require looking for colocation of Eurasian + other sub-Saharan haplotypes (passed along together in a single tract that itself was the product of a previous admixture + recombination). Such a signal would indeed be strong evidence of Eurasian haplotypes entering via a previous admixture, but as far as I can tell GLOBETROTTER only looks at copying vectors (vs. relative locations of copied segments).

---

## [Author Response]

*Essential revisions:*

*The reviewers raised a number of important points, specifically regarding the interpretation of the analysis. There should be some additional attention paid to these points at the relevant spots in the manuscript. I (the editor) have lightly edited some of the major points raised by the reviewers below.*

1) Reviewer #2 argued that the results of the analysis are highly dependent upon the available samples. While the sampling here is better than previously available, relative to the actual scale of Africa it remains under sampled in several key regions. This fact is acknowledged at several points in the manuscript, however, much of the interpretation of the results are done without considering how sampling may influence their findings. For example, a central focus of the manuscript is on the Bantu expansion, however there are no samples included from the purported area of the Western branch of their spread south. This is a major limitation that needs to be considered at all points.

In the original manuscript we aimed to stress the limitations of our dataset in relation to the lack of representation from autochthonous hunter-gatherers from outside of southern Africa. We admit that there are other areas of the continent where we do not have samples, and that this will have affected our interpretation of events. Following this, and other, comments by Reviewer #2 regarding our interpretation of the Bantu expansion, we agree that there is a need to stress that we do not have a good representation of the Western Bantu from countries such as Gabon, Congo, DRC, and Angola.

Please see our response to the next comment for how we propose to account for this.

*2) Reviewer #2 argued that the authors' reading of the available literature on African prehistory is a bit thin at times. These analyses will be much more valuable and have a much wider impact if they are better integrated into the reasonably well developed archaeological and linguistic understanding of African population history*

The primary disconnect from the interpretations presented here and the existing literature is with respect to the Bantu expansion. The authors consider their Cameroon samples to best represent the western branch of the Bantu migration. However, the archaeological and linguistic consensus seems to be that the area of Cameroon is actually the origin of the Bantu expansion. In other words, the Cameroon samples may best represent something like the ancestral population of both the Eastern and Western branches of the expansion. Also, since the Cameroonian samples are the most southern and western samples included from the West African set, it means that, contrary to the author's assertions, no western branch Bantu populations were included in this study. Considering the Cameroonian samples as representative of the origin, rather than the western part of the expansion will change the interpretation considerably.

Thank you for this and other comments regarding interpretation of the results. Several of the revisions suggested by the reviewers address our interpretation of the Bantu expansion. We will endeavour to outline here the changes that we have made regarding these comments in line with the next few comments below. In response to comments (2)-(7), unless otherwise stated, we will refer exclusively to the section of the Results titled “Population movements with Africa and the Bantu expansion”.

We agree that the Cameroon samples (from the Bantu and Semi-Bantu ethnic groups) are likely to be the best representatives in our dataset of the ancestral group of both the Western and Eastern branches of the Bantu expansion. We wrote: “Assuming that Cameroon populations are the best proxy for Bantu ancestry in our dataset”, but this appears to have led to confusion. In the changes outlined below, we have highlighted our logic behind this further, in particular in the fifth paragraph of the subsection “Population movements within Africa and the Bantu expansion”.

We do have one member of the western Bantu. The Herero from Namibia speak a language belonging to the ‘R’ subgroup of the Central Bantu languages, which Guthrie (1948) and Mace (2002) both classify as Western Bantu, the latter based on a phylogenetic analysis of 75 Bantu and Bantoid languages. It could also be argued that the Niger-Congo ancestry in the Khoesan groups from Namibia and Angola, which was previously described by Schlebush et al. (2012), likely came from interactions between local groups and Western Bantu speakers.

In our re-write of the subsection titled “Population movements with Africa and the Bantu expansion” we have described more precisely our dataset, making explicit the lack of western Bantu speakers other than the Herero (fifth paragraph). We have also described the caveats in which our interpretation should be contextualised. Finally, we have described the logic behind why we still believe that our analysis allows us to comment on the direction of the Bantu expansion. In particular, in the seventh paragraph of the aforementioned subsection, we have described the restricted surrogate analysis in more detail, and why we think that this analysis can provide some clarity on the direction and timing of the Bantu expansion.

3) MALDER seems to give consistently older dates than GLOBETROTTER? Why? Which should we prefer? The only way to properly address this question is to attempt to relate the inferred dates to the known historical or archaeological record (e.g. dates that are independent of population genetic inference). Some contextualisation here would be extremely helpful. Otherwise there is no clear way for the reader to evaluate these findings.

We compare dates inferred by MALDER and GLOBETROTTER in Figure 4. In our first submission, we incorrectly referred to Figure 3 and Figure 3 in the Methods section “3.12 Comparing weighted LD curve dates with GLOBETROTTER dates”, which has now been changed. We can see from the two Figure 4 panels (Figure 4) that the two dating methods give broadly similar dates, with a few exceptions.

Note that we have now altered the date plotting of the main panel (A) in Figure 3 to more closely resemble the dates shown in Figure 4. Specifically, we have (1) converted the date confidence intervals (CI) from +/- 1 standard error (SE) to +/- 1.96 SE to be more closely comparative to the 95% CIs generated by GLOBETROTTER, and (2) we have put dotted vertical lines at 1000 year intervals. We hope that it is now clearer that there is no systematic difference between the dating results.

4) In the second paragraph of the subsection “Population movements within Africa and the Bantu expansion”. Tishkoff et al. 2009 showed pretty strongly convincingly using genomic STR markers that the Bantu expansion involved a spread of West African people, rather than simple cultural diffusion. Thus, the characterisation of this as 1) an ongoing controversy, and 2) one only addressed using uniparental markers is not correct. In fact, Tishkoff et al. did so with even greater population sampling than found in the present study.

We wrote: “Whether which this cultural [Bantu] expansion was accompanied by people is an active research question”. This is true: we cite recent research on the genetics of the Bantu expansion published within the last two years by Li et al. (2014) and Gonzales-Santos et al. (2015), as well as other work published with the last 15 years. We do not mean this to imply that there is an “ongoing controversy”, but see how it could have been read that way. We do think that it is a subject that has not been extensively studied with autosomal data, and in particular not using the latest admixture inference techniques. However, to address the reviewer’s concerns, we have rephrased this sentence and moved it to the Introduction:

“The extent to which this cultural expansion was accompanied by people is an active research question, but an increasing number of molecular studies indicate that the expansion of languages was accompanied by the diffusion of people [Beleza et al., 2005; Berniell-Lee et al., 2009; Tishkoff et al., 2009; Pakendorf et al., 2011; de Filippo et al., 2012; Ansari Pour et al., 2013; Li et al., 2014; Gonz´alez-Santos et al., 2015].”

A separate issue is whether the Tishkoff et al. (2009) analysis “convincingly” demonstrated that the Bantu expansion involved a spread of West Africa people. Tishkoff et al. (2009) used autosomal microsatellite data, analysed with the STRUCTURE clustering algorithm and TESS (a STRUCTURE-like model that incorporates spatial information), to demonstrate the presence of a geographically widespread cluster from Western to Southern Africa. They present results from various iterations of the STRUCTURE algorithm where they alter the value of K, the prescribed number of ancestral groups. This analysis shows that there is shared ancestry across much of sub-Saharan Africa, but doesn’t explicitly test whether this is the result of the Bantu expansion (something that Li et al. (2014) do attempt with the same dataset). Moreover, as stated in the original STRUCTURE paper (Pritchard et al. 2000), “clusters may not correspond to real populations”, so it is not possible to say, from this analysis alone, that the observed patterns are the result of the Bantu expansion. In the current manuscript, we hope to have added to the debate by providing a fuller characterisation of the sources and timings of admixture involving “Bantu-like” groups at archaeologically acceptable dates. (See next comment.)

5) In the second paragraph of the subsection “Population movements within Africa and the Bantu expansion”. The descriptions of the two linguistic hypotheses are particularly lacking in detail. It is important to put dates on the possible early and late splits, since the magnitude of the difference between these will determine the power to detect a difference genetically. Also, it is stated that recent genetic and linguistic analyses support the late split, but no mention of how. This is important for contextualising how this analysis adds to these other findings.

We agree that more description is needed here, and specifically welcome the suggestion that we should include dates to the dispersals. We hope that the revision of this section better characterises the competing hypotheses (subsection “Population movements within Africa and the Bantu expansion”, third and fourth paragraphs), and we have tried to clearly lay out the arguments for the two hypotheses.

6) In the third paragraph of the subsection “Population movements within Africa and the Bantu expansion”. I do not understand the logic behind the claim that the early split hypothesis predicts all of the ancestry of Eastern NC speakers comes from Central West Africa. The general consensus is that the Bantu expansion originated in the area of Cameroon or Nigeria ~5kya, then split into two major forks (Greenberg 1972, Phillipson 1975). The first spread around the Congo forest to the North, and entered East Africa in the Great Lakes region and continued down the eastern margin of Africa to Southern Africa. The second went around the Congo Forest down the Western Coast and entered Southern Africa. These two branches are thought to have ultimately reunited in Southern Africa. The analysis presented here finding Southern African NC ancestry in East African NC speakers might best be explained by the former descending from the latter. The ability to detect ancestry from western branch of the Bantu expansion is not possible here because there are no populations sampled from the area of Gabon, Angola, etc. I am worried these findings are a sampling artefact.

If (as acknowledged by the reviewers) we can assume that our Cameroon populations (part of the Central West African NC ancestry region) represent the best proxy for the ancestral Bantus, then we can use this to test the two split hypotheses. If East African NC speakers were the result of an early eastward migration across the top of the equatorial forests, then we’d expect to observe an old signal of admixture involving Cameroon surrogates in the admixture sources of these groups. If East African NC speakers were the result of a later split from the western Bantu instead, then we’d expect them to be more closely related to contemporary groups close this split, i.e. from south central Africa. To describe this more concisely, we now write:

“The current dataset does not cover all of Africa. In particular, it contains no hunter-gather groups outside of southern Africa, and no representation of the western Bantu except the Herero from Namibia. […] We note that these may change with future analyses involving populations from the relatively under-sampled central southern Africa.”

What we observe in East Africa NC speakers are events involving a major source containing surrogates most similar to Malawi (Southern Africa NC; Figure 4 and supplements) within the last 2,000 years. We believe this to be more consistent with the late-split hypothesis, as it suggests that the admixture events 2,000 years ago involved southerly Bantu groups, and not groups from Cameroon. As the reviewer suggests, an alternative explanation is that we see this relationship because the southerly Bantu groups are themselves descended from the East African NC speakers. However, we believe that we can get at the directionality of movement, with notable caveats, with our restricted analysis. In an effort to better describe our logic in relation to the restricted surrogate analysis, we have revised the relevant paragraphs (subsection “Population movements within Africa and the Bantu expansion”, seventh and eighth paragraphs).

7) In the subsection “(3) Medieval contact between Asia and the East African Swahili Coast”. Again it is not clear that this represents the Western Bantu expansion since the samples come from Cameroon which is hypothesised to be near the homeland of both the Eastern and Western branches of the expansion.

We hope that we have provided sufficient additional detail and caveats in the section re-write, and in answer to comment (56) above to enable us to postulate that the Central West African ancestry observed in the south-west African groups is the result of gene-flow through the western Bantu.

8) In the subsection “(7) Pre-Bantu pastoralist movements from East to South Africa”. The proposed model here is incongruent with the known archaeology, which shows the Bantu expansion reaching the eastern side of the Congo Forest in the north near Lake Victoria, rather than the south. The archaeology may not be correct, however given the lack of western Bantu samples included in this study, the conservative interpretation is that these findings are a sampling artefact.

As we have hopefully described more fully in the revised manuscript, the model that the reviewer suggests, of an early expansion to eastern Africa north of the Congo forest to Lake Victoria, is referred to as the early-split hypothesis. We believe our analysis, in addition to other recent genetic (Li et al. 2014) and linguistic (Grollemund et al. 2015) work is more consistent with the late-split hypothesis. We wrote at the beginning of this subsection (“A haplotype-based model of gene-flow in sub-Saharan Africa”) that “Using genetics to infer historical demography will always depend on the available samples and population genetics methods used to infer population relationships.” We therefore hope that we have outlined the relevant caveats for our model of gene-flow events in both this section, and the previous one, regarding the Bantu expansion.

9) In the first paragraph of the Discussion. The adoption of pastoralism began well before 2,000 years ago. The earliest archaeological evidence of cattle keeping may date to as early as 9kya in the Nile River Valley, and is well established by 6kya (see Boivin et al. 2010). Pastoralists reach the Koobi Fora region of Kenya by 4.5 kya, and are east of Lake Victoria by 3.8 kyat (see Pendergast 2011).

Thank you for this comment. We have altered this sentence to read:

“An unanswered question in African history is how contemporary populations relate to those present in Africa before the transition to pastoralism that began some 9kya.”

10) In the first paragraph of the Discussion. Why is the GLOBETROTTER analysis more precise? Is there any evidence to support the assertion? Also, even if more precise, is it more accurate?

This is a statement about how well the sources of admixture are characterised by the two admixture inference approaches. In Figure 3 we show that often all we can say about the sources of admixture using MALDER is that they contain ancestry from a set of (usually multiple) ancestry regions. GLOBETROTTER on the other hand allows us to characterise how the different admixture sources are composed, as well as the proportion of admixture coming from each source.

The precision of GLOBETROTTER was tested against ROLLOF (an earlier incarnation of MALDER) in the original GLOBETROTTER paper (Hellenthal et al. 2014) and found to provide date estimates with tighter confidence intervals on simulated data than ROLLOF. Accuracy was also assessed in that paper, with the confidence interval containing the true date in over 90% of ~5,000 simulated admixture scenarios.

However, we note that precision has a particular scientific meaning, which has not been explicitly tested in the current setting. We therefore removed this sentence from the Discussion, and instead write:

“The f3 and MALDER analyses show evidence for deep Eurasian and some hunter-gatherer ancestry across Africa, to which our GLOBETROTTER analysis (Figure 4) provides further clarity on the composition of the admixture sources, as well as the timing of events and their impact on groups in our analysis (Figure 6).”

11) Methods section: The Methods section is often difficult to follow because it frequently mixes results with the methods themselves. If results of one analysis are necessary to inform how the following analysis is to be performed, it would be preferable to lay out the strict methods and the logic of how methodological options were chosen in the Methods section, and then keep the actual progression of results in the Results section.

Thank you for this comment. We have edited the Methods section to keep to the suggested logical progression. In particular we:

A) Integrated the first paragraph of section 2.4 describing the fineSTRUCTURE results to the Results section;

B) Removed the description of the MALDER curve amplitude analysis from section 3.1;

C) We renamed Section 3.5 and moved text to the figure supplement legends;

D) We renamed Section 3.6;

E) We moved the section (3.15) titled “Summary of inferred gene-flow from West to East and South Africa” to a new [Supplementary-material SD14-data]. This avoids adding more text to the main Results section of the manuscript.

12) More generally, the manuscript is a bit too long; in particular, the Discussion and Abstract contain a substantial amount of content on genetic epidemiology and infectious disease that is only tangentially related to the analyses performed in the manuscript.

We have chosen to provide context to our analysis, both in relation to previous work by using similar admixture inference methods to previous authors together with our “new” haplotype-based analyses of African genetic variation, but also by discussing more generally how results such as ours should be interpreted (“Interpreting haplotype similarity as historical admixture”), as well as the relevance of this work to the wider field (“Admixture and genetic epidemiology”).

We have made an effort to trim the manuscript to improve clarity whilst hopefully retaining important context and caveats (particularly the Introduction and Discussion). We have tried to keep to the ethos of *eLife* by providing minimal supplementary information with the manuscript and putting all of our content into the main manuscript, which, in addition to the description and discussion of the multiple different analyses has led to a long paper.

13) The description of the TVD analyses (subsection “Haplotypes reveal subtle population structure”) could be clarified. The comparison of TVD vs. Fst suggests that TVD is more informative ("expected to capture more variability") because recombination rates are faster than mutation rates. However, aren't recombination rates and mutation rates quite similar (both roughly 10^-8 per base per generation)? Also, the difference between TVD and Fst seems to be more than just amount of information captured; the two metrics must be capturing different types of information given the empirical differences between the values of the metrics. More intuition would help. Overall, I found the discussion of TVD rather confusing, especially given that paragraph ends with a caveat about interpretation – what then should we make of the TVD results in Figure 2?

F_ST and TVD are both metrics that allow one to generate a single value that represents the genetic difference between two populations. Whilst F_ST is based on the variance in allele frequencies between populations, TVD is measures the absolute differences in copying between two populations by comparing their copying vectors. The reviewer is right to say that they are capturing different types of information: F_ST is a measure of how drifted two populations have become, whereas TVD attempts to measure shared ancestry directly. Another way of thinking about it is that TVD reflects similarities across sets of contiguous SNPs, or haplotypes, whereas F_ST looks at each allele independently. We have rephrased the section on TVD to better describe the metric and the intuition about what it can tell us.

We have also added an additional figure (Figure 2—figure supplement 1) where we provide an additional comparison of a genotype-based analysis (PCA) and fineSTRUCTURE, which uses haplotypes. We hope that this adds to the aim of this section, which is to outline our motivation for using haplotype-based methods as an alternative way of inferring relationships between populations.

*14) In the main GLOBETROTTER analyses, the authors note that the GLOBETROTTER approach allows them to infer whether Eurasian haplotypes came directly into sub-Saharan Africa or were brought indirectly together with sub-Saharan groups (subsection “Direct and indirect gene flow from Eurasia back into Africa”, second paragraph). Is this statement technically correct? It seems to me that inference of this sort would require looking for colocation of Eurasian + other sub-Saharan haplotypes (passed along together in a single tract that itself was the product of a previous admixture + recombination). Such a signal would indeed be strong evidence of Eurasian haplotypes entering via a previous admixture, but as far as I can tell GLOBETROTTER only looks at copying vectors (vs. relative locations of copied segments).*

This statement is based on the idea that GLOBETROTTER can reconstruct the genetic composition of admixture sources as mixtures themselves. GLOBETROTTER does this by identifying whether ancestry chunks from two different populations (as inferred through chromosome painting) tend to be situated at close genetic distances, in which case ancestry from both populations will have arrived together, or whether these ancestry chunks tend not to be associated with each other, in which case they likely arrived from different admixture source inputs. We use the painted chromosomes to look at the distribution of these ancestry chunks, and so the mixtures in these source groups are indeed based, in some sense, on the co-localisation of ancestry in the genome. The source group composition is thus informative of the identity of the admixture sources, so when we observe an admixture source which is almost exclusively Eurasian, then this does imply that the admixture source (given the available surrogates used) was likely to have been mostly Eurasian.